

# An International Intercomparison of Continuous Flow Analysis (CFA) Systems for High-Resolution Water Isotope Measurements in Ice Cores

Agnese Petteni[1,2], Elise Fourré[2], Elsa Gautier[3], Azzurra Spagnesi[4,1], Roxanne Jacob[2], Pete D. Akers[3,5], Daniele Zannoni[1], Jacopo Gabrieli[4], Olivier Jossoud[2], Frédéric Prié[2], Amaëlle Landais[2], Titouan Tcheng[2], Barbara Stenni[1], Joel Savarino[3], Patrick

Ginot[3], Mathieu Casado[2]

[1]Ca' Foscari of Venice, Department of Environmental Sciences, Informatics and Statistics, Mestre (Venice), Italy
[2]LSCE/IPSL, CEA-CNRS-UVSQ, Université Paris-Saclay, Gif-sur-Yvette, France
[3]Université Grenoble Alpes, CNRS, IRD, Grenoble INP, INRAE, IGE, F-38000 Grenoble, France
[4]Institute of Polar Sciences, National Research Council of Italy, Venice, Italy
[5]Geography, School of Natural Sciences, Trinity College Dublin, Ireland

*Correspondence to*: Agnese Petteni (agnese.petteni@unive.it)

**Abstract.** The Continuous Flow Analysis technique coupled with Cavity Ring Down Spectrometry (CFA-CRDS) provides a method for high-resolution water isotope analysis of ice cores, which is essential for paleoclimatic reconstructions of local temperatures and regional atmospheric circulation. Compared to the traditional discrete method, CFA-CRDS significantly

reduces analysis time. However, the effective resolution at which the isotopic signal can be retrieved from continuous measurements is influenced by system-induced mixing, which smooths the isotopic signal, and by measurement noise, which can further limit the resolution of the continuous record introducing random fluctuations in the instrument' signal output. This study compares three CFA-CRDS systems developed at CNR ISP-Ca' Foscari University (Venice), the Laboratoire des Sciences du Climat et de l'Environnement (Paris), and the Institut des Géosciences de l'Environnement (Grenoble) for firn core

analysis. Continuous results are compared with discrete data to highlight the strengths and limitations of each system. A spectral analysis is also performed to quantify the impact of internal mixing on signal integrity and to determine the frequency limits imposed by measurement noise. These findings establish the effective resolution limits for retrieving isotopic signals from firn cores. Finally, we discuss critical system configurations and procedural optimizations that enhance the accuracy and resolution of water isotope analysis in ice cores.

## 1 Introduction

Water isotopes are valuable proxies for studying past climatic processes, providing insights into local temperature and atmospheric circulation patterns (Dansgaard, 1964; Petit et al., 1999). In permanently frozen regions of Antarctica, the glacial ice sampled through ice coring serves as a continuous archive of past climate conditions, with annual snowfall creating new layers at the top of the ice sheet every year. Low-accumulation areas like the East Antarctic Plateau (20-50 mm weq yr$^{-1}$) often

yield records with decadal or multi-decadal resolution (Casado et al., 2020; Petit, 1982). By contrast, in high-accumulation areas (100-300 mm weq yr$^{-1}$), such as coastal regions in East Antarctica and West Antarctica, ice core records can achieve



seasonal or annual resolution (Alley, 2000; Markle et al., 2017; Sigl et al., 2016). Depositional processes, such as precipitation intermittency (Casado et al., 2020; Laepple et al., 2011; Steig, 1994), introduces bias into the snow layers' recorded signal. In addition, post-depositional processes, including wind-driven snow redistribution (Picard et al., 2019), sublimation, and

condensation, introduce stratigraphic noise into the snow's surface composition and alter local surface values (Casado et al., 2021; Wahl, 2022). Meanwhile, isotopic composition below the surface can be modified over time through processes like diffusion (Genthon et al., 2017). The deformation of deeper ice layers under the weight of the ice sheet compresses the timescale retrieved from each centimetre of ice analysed (Huybrechts et al., 2007). Therefore, high-resolution analyses are critical for ice cores to preserve the integrity of the isotopic signal. While deep ice cores from low-accumulation areas are

traditionally analysed using discrete sampling with resolution ranging from 10 cm to 50 cm (Grisart et al., 2022; Landais & Stenni, 2021), higher resolution (< 10 cm) is achieved in cores from high-accumulation areas, where seasonal signal can be detected (Crotti et al., 2022; Goursaud et al., 2017). In both cases, although precise, this method is time-consuming due to the extensive sample preparation and the need for offline analysis. In contrast, Continuous Flow Analysis (CFA) coupled with a Cavity Ring-Down Spectrometer (CRDS) has emerged as a more efficient alternative, enabling high-resolution measurements

of water isotopes (Gkinis et al., 2010). This system slowly and continuously melts solid ice sticks at the base, with the resulting liquid water directed into a vaporiser before being injected into a CRDS instrument - typically a Picarro-brand isotopic analyser. This method eliminates the need for manual sample handling, significantly reducing analysis time. For instance, within the East Antarctic International Ice Sheet Traverse (EAIIST - Traversa et al., 2023; Ventisette et al., 2023) project, the analysis of 18 firn cores collected on the Antarctic Plateau (total length of about 960 m) would results in analysing

approximately 20,000 samples at resolution of 5 cm. This would take more than two years of discrete analysis if conducted with a single CRDS instrument. In contrast, with CFA-CRDS, operating at a melt rate of 2.5-3 cm min$^{-1}$, the same analysis can be completed in roughly three months, with the capability to process up to 10 meters of ice core per day. Despite its advantages, CFA-CRDS faces several technical limitations, one being the mixing of water molecules within the system, leading to signal smoothing (Gkinis et al., 2011). This mixing can occur at various stages, from the melt-head to the instrument

cavity, displacing water molecules from their original relative positions. The extend of this effect can vary with changes in melt rate, especially at the melt-head level due to capillary action. As a result, isotopic values must be averaged over a depth range corresponding to the mixing length to accurately represent the preserved climatic signal (Gkinis et al., 2011; Jones et al., 2017a). Additionally, measurement noise – referring to random fluctuations in the instrument' signal output – further limits the effective resolution, restricting the ability to retrieve meaningful climatic signals at high frequencies. Accurately

determining the delay time between the melt-head and CRDS signals is critical for converting the timescale to a depth scale. Additionally, issues during core melting, such as temporary blockages or stick collapses, introduce uncertainties when assigning depth to the isotopic profile.

In this study, we present the 4-m section of the PALEO2 core (EAIIST), analysed at three research institutes: CNR ISP-Ca' Foscari University (ISP-UNIVE - Venice, Italy), Laboratoire des Sciences du Climat et de l'Environnement (LSCE - Paris,

France), and the Institut des Géosciences de l'Environnement (IGE - Grenoble, France). These laboratories collaborate on



international ice core projects such as EAIIST and Beyond EPICA Oldest Ice Core (BE-OIC - Lilien et al., 2021; Parrenin et al., 2017) and share common samples, emphasising the need for accurate comparisons between the different systems. We compared CFA results with discrete profile at 1.5 cm resolution to highlight the strengths and limitations of each technical setup and operating procedure. Power spectral density (PSD) analysis is used to quantify the effects of mixing on the signal and to determine the maximum resolution achievable by each system, as constrained by internal mixing and measurement noise.

## 2 Methods

### 2.1 Ice Core Processing

Four ice cores were drilled at the PALEO site (79°64'S, 126°13'E) during the EAIIST on the Antarctic Plateau (Traversa et al., 2023; Ventisette et al., 2023). Here, we use the PALEO2 firn core (18 m) to compare three CFA-CRDS systems. The entire core was continuously analysed at LSCE in June 2023, while 4-m sections (depth 12-16 m) of the same core were analysed at IGE in July 2023 and at ISP-UNIVE in January 2024. This depth interval, with a density of about 0.58 g cm$^{-3}$, was chosen to provide new insights into the analysis of low-density sections of the cores, while ensuring sufficient structural integrity during processing in the cold room. The core was cut into about 1.00x0.03x0.03 m sticks (Fig. 1), with cuts for LSCE and IGE sticks done in June 2023 and for ISP-UNIVE in January 2024. During the preparation of the ice sticks for the ISP-UNIVE measurements, we also prepared discrete samples with a depth resolution of 1.5 cm, which were stored frozen in PTFE bottles and analysed offline. Spectral analyses (Sect. 3.3, Fig. 8) show no significant diffusion occurred during the six-month storage at -20°C between the two cutting sessions. The remaining quarter of the core was stored in plastic bags for any future investigation.

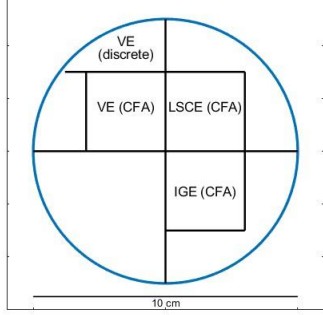

**Figure 1: Cutting scheme of PALEO2 core for CFA-CRDS systems and discrete sampling.**



## 2.2 Isotopic Measurements and Data Calibration

The isotope values of $\delta D$ (or $\delta^2H$) and $\delta^{18}O$ are here reported as the deviation of the ratio of heavy to light isotopes relative to the Vienna Standard Mean Ocean Water (V-SMOW) international standard, where $\delta$ values are expressed in parts per mil (‰):

$\delta(‰) = (R_{SAMPLE}/R_{VSMOW}-1)*1000$

where $R_{SAMPLE}$ and $R_{VSMOW}$ are the ratios between $^{18}O$ /$^{16}O$ or D/H in the sample and in V-SMOW, respectively. The second-order parameter deuterium excess ($d_{xs}$) is defined as follows (Craig, 1963; Dansgaard, 1964):

$d_{xs} = \delta D - 8*(\delta^{18}O)$

All reported isotope values are calibrated against the internal laboratory standards (STDs) provided by each institute (Tab. 1), which are in turn calibrated against international reference waters V-SMOW and SLAP (Standard Light Antarctic Precipitation). Calibration involves applying a linear regression between the "measured" and "known" values of STDs, using the resulting slope and intercept to correct the sample data to the reference scale. Calibration STDs have isotopic compositions similar to those expected for Antarctic ice cores (Landais & Stenni, 2021) to minimize the instrument's memory effect and accurately capture the highly negative isotopic values typical of polar snow. At ISP-UNIVE, a Picarro L2140-i was utilized for the discrete analysis. The standards TD (Talos Dome) and AP1 (Antarctic Plateau 1) were used for calibration, while two vials of DCS (Dome C Snow) were analysed as controls. The accuracy of offline measurements was determined as the mean difference between control and true values of the STDs controls, with uncertainty represented by their SD. This yielding an accuracy of -0.01‰ for $\delta^{18}O$, -0.07‰ for $\delta D$, and -0.02‰ for d-excess, with corresponding uncertainties of ±0.07‰, ±0.4‰, and ±0.4‰, respectively. The three CFA systems discussed in this study are all coupled with CRDS Picarro-brand isotopic analysers. Continuous raw data are calibrated to the V-SMOW/SLAP similar to discrete approach, but injecting water STDs at a constant humidity level, similar to that of continuous analysis, before and/or after each daily measurement. At ISP-UNIVE laboratory, calibration standards (AP1 and DCS) are injected for 20 min, while the last 5 min are selected to minimise the SD. At LSCE, standards (NEEM, Adelie, and OC4) are injected for 25 min selecting the final 3 min. At IGE, the standards (EDC, EGRIP-01, and SOUP-01) are injected for 10 min and a 3-min interval is selected. At the ISP-UNIVE, STDs are measured at the start of the day, while at LSCE and IGE, STDs are measured both at the beginning and end of the analysis. At LSCE, three calibration methods - using start-day, end-day, or the average of both - were tested, showing no significant measurement drift throughout the day.

**Table 1: Internal laboratory isotopic STDs values (‰) used for the V-SMOW/SLAP calibration at a) ISP-UNIVE, b) LSCE and c) IGE.**

|  | Standards | $\delta D$ (‰) | $\delta D$ uncertainty (‰) | $\delta^{18}O$ (‰) | $\delta^{18}O$ uncertainty (‰) |
|---|---|---|---|---|---|
| **a) ISP-UNIVE** | TD | -304.9 | 0.7 | -38.36 | 0.10 |
|  | DCS | -407.4 | 0.7 | -51.95 | 0.10 |
|  | AP1 | -424.2 | 0.4 | -54.56 | 0.07 |
| **b) LSCE** | NEEM | -254.1 | 0.7 | -32.89 | 0.05 |
|  | Adelie | -321.0 | 0.7 | -40.55 | 0.05 |



| | | | | | |
|---|---|---|---|---|---|
| | OC4 | -422.7 | 0.7 | -53.93 | 0.05 |
| c) IGE | EGRIP-01 | -281.0 | 0.1 | -36.34 | 0.01 |
| | GREEN-01 | -383.8 | 0.5 | -48.97 | 0.03 |
| | SOUP-01 | -388.1 | 0.1 | -49.48 | 0.01 |


## 2.3 CFA-CRDS Coupled System

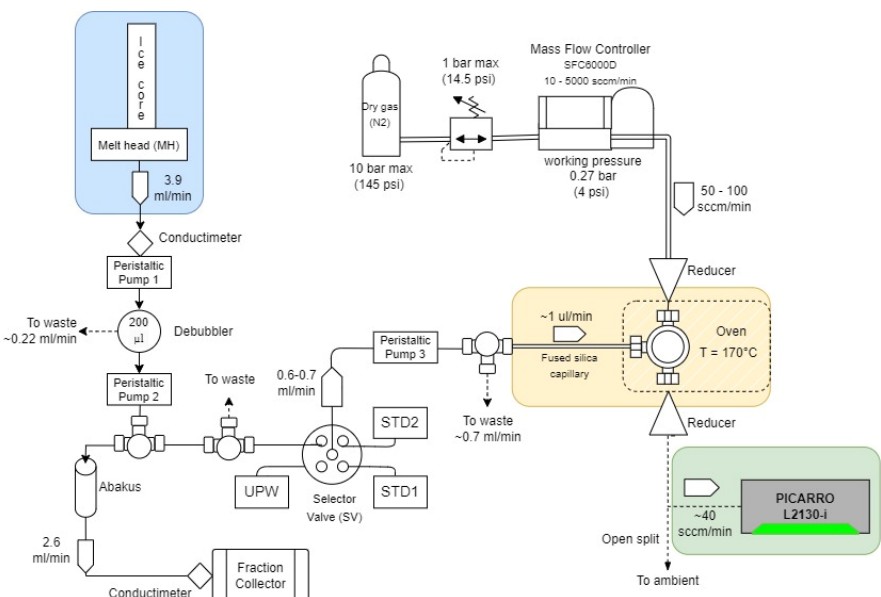

**Figure 2: Schematic of the CFA-CRDS system at ISP-UNIVE. The ice core-melting component, the vaporizer and the isotopic analyser are highlighted in blue, yellow and green, respectively. The melting rate is equal to 3 cm min⁻¹.**

The CFA-CRDS systems are commonly composed of three main units (Jones et al., 2017a): the ice core-melting component located in a cold room, which generates continuous stream of liquid water (Fig. 2 – blue block), a vaporiser that converts the liquid into vapor (Fig. 2 – yellow block), and the CRDS isotopic analyser (Fig. 2 – green block). This setup enables online water isotope measurements of $\delta$D and $\delta^{18}$O during the melting of the cores, which are cut in sticks and are loaded vertically above the melt-head (MH). The transport line typically includes a debubbler, which permits the release of air bubbles from the

water stream as it passes through, but also promotes additional mixing. A selector valve (SV) prior to the vaporiser allows switching between the CFA line, Ultra Pure Water (UPW) and calibration STDs. The water vapor is transported from the vaporiser to the analyser using a carrier gas ($N_2$ or zero air). The key differences between the three laboratories are summarised in Tab. 2. The novel ISP-UNIVE-CFA system (Fig. 2) is highly scalable according to the specific laboratory needs (Barbaro et al., 2022; Spagnesi et al., 2023). The melting unit is located within a vertical freezer. A conductivity device monitors the

meltwater stream prior it enters the small-volume 200 µl debubbler, which regulates the overflow by continuously discarding



part of the flow (~0.22 ml min⁻¹). The isotope line flows through an open T before reaching the vaporiser, possibly promoting ambient air intrusion into the Picarro L2130-i analyser in case of stick blockage. The vapor mixing ratio is kept between 10,000-14,000 ppmv. The LSCE-CFA system (Appendix A, Fig. A1) is dedicated to continuous isotope measurement and discrete sampling. A 430 µl debubbler is regulated by an automatic flow control to prevent overflow, and the system includes

a dust filter (A-107 IDEX stainless steel filter, 10 µm size: .189" x .074" x .254") and three conductivity devices. It keeps vapor mixing ratios between 18,000-22,000 ppmv. The IGE-CFA system (Appendix A, Fig. A2) supports a wider range of online analyses, featuring a 1,000 µl debubbler where the flow is manually regulated by pump adjustment. Meltwater passes through a 180 µm dust filter and a longer distribution line. The isotopic line includes an additional filter (identical to the LSCE one) and a conductivity device prior the analyser, maintaining a humidity levels between 17,000-21,000 ppmv.

**Table 2: Technical properties for different CFA-CRDS setups**

| CFA-CRDS setup | ISP-UNIVE | LSCE | IGE |
|---|---|---|---|
| **Melt rate (cm min⁻¹)** | 3.0±0.5 | 2.5±0.5 | 3.0±0.5 |
| **Online analysis performed** | Water isotopes, insoluble particles | Water isotopes | Water isotopes, ICPMS, insoluble particles, black carbon, TOC and colorimetry |
| **Conductivity devices for isotope line** | 1 | 3 | 1 |
| **Pumps for isotope line** | 3 | 2 | 2 |
| **Debubbler volume (µl)** | 200 | 430 | 1,000 |
| **Filter** | No | 10 µm filter for the isotopic line | I) 180 µm filter for the main line II) 10 µm filter for the isotopic line |
| **CRDS instrument** | Picarro L2130-i | Picarro L2130-i | Picarro L2140-i |
| **Flow rates to the Picarro (ml min⁻¹)** | 0.6-0.7 | 0.6-0.7 | 0.5-0.7 |
| **Humidity (ppmv)** | 10,000-14,000 | 18,000-22,000 | 17,000-21,000 |

## 2.4 CFA-CRDS Systems Performance

### 2.4.1 Impact of the Humidity Level

The impact of humidity on the isotopic measurements was evaluated for the three water vapor mixing ratio ranges used during the analyses. The range maintained at ISP-UNIVE, between 10,000 and 14,000 ppmv with occasional fluctuations down to

8,000 ppmv, was assessed using laboratory standard AP1, analyzed in 5-minute intervals at steps of 8,000, 9,500, 11,500, and 14,000 ppmv. For each step, the last 3 minutes were selected. The differences in $\delta^{18}O$ and $\delta D$ between these humidity levels were smaller than the Allan Deviations (see Sect. 2.4.2) for a 1-second integration time, corresponding at the resolution of the data Picarro output (Fig. 3). For LSCE setup, the same approach was followed with humidity steps performed between 17,000 and 23,000 ppmv. For IGE setup, we relied on above tests confirmed the findings of Gkinis et al., (2010) who validated that

variations in $\delta^{18}O$ and $\delta D$ can be neglected at water vapor mixing ratio in the range 15,000-22,000 ppmv. Consequently, we opted to not apply humidity level correction to the data.



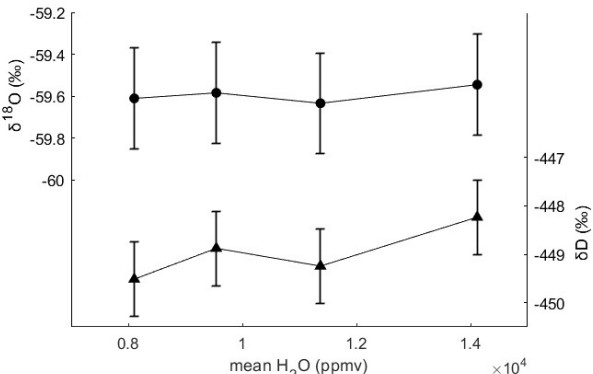

**Figure 3: Mean $\delta^{18}$O and $\delta$D values for the 3-minute intervals selected at humidity level of 8,000, 9,500, 11,500 and 14,000 ppmv in the ISP-UNIVE setup. The confidence levels are defined as the Allan Deviation for integration time of 1 s (±1σ).**

### 2.4.2 Continuous Measurements Noise

Measurement noise is quantified using Allan Variance (Allan, 1966) from continuous measurement of a sample of constant isotopic value under humidity conditions that match those CFA-CRDS analyses. Allan Deviation, is an indicator of the stability of the vapor phase signal across varying integration times (τ). The calculation is based on at least 2 hours of UPW flow into the CRDS analyser under stable humidity conditions (Fig. 4, Tab. 3) and thus reflects the stability of the combined vaporiser and CRDS analyser. For the ISP-UNIVE, LSCE, and IGE setups, we select a 1-hour interval with mixing ratios of 14,800±53 ppmv, 20,380±190 ppmv, and 18,100±105 ppmv, respectively. All systems show a decrease with a slope of $N^{-1/2}$, characteristic of white noise, for at least τ=250 s, indicating that precision improves with longer integration times. For τ=2 s (corresponding to melting 0.1 cm of ice at 3.0 cm min$^{-1}$), the SD for $\delta^{18}$O are 0.17‰ (ISP-UNIVE), 0.15‰ (LSCE), and 0.06‰ (IGE). At τ=30 s (time required to melt 1.5 cm of sample), precision improves, with SDs decreasing to 0.05‰, 0.04‰, and 0.02‰, respectively. The IGE system exhibits higher precision, attributed to better instrument performance as indicated by Allan Deviation and the elevated working humidity levels. At ISP-UNIVE, the humidity range of 10,000-14,000 ppmv is used despite being outside the instrument's optimum, as large water volumes are required for discrete chemical analysis, and the N$_2$ flow rate allows for rapid adjustments in case of decreases in the water flow.

**Table 3: Allan Deviation for $\delta^{18}$O, $\delta$D and d-excess concerning the integration times (τ) needed to melt 0.1 cm and 1.5 cm of the ice core at the nominal melting rate for each institute.**

|  | Melt rate | SD $_{0.1\ cm}$ | | | SD $_{1.5\ cm}$ | | |
|---|---|---|---|---|---|---|---|
|  | (cm min$^{-1}$) | $\delta^{18}$O (‰) | $\delta$D (‰) | d$_{xs}$ (‰) | $\delta^{18}$O (‰) | $\delta$D (‰) | d$_{xs}$ (‰) |
| **ISP-UNIVE** | 3.0 | 0.17 | 0.55 | 1.47 | 0.05 | 0.16 | 0.38 |
| **LSCE** | 2.5 | 0.15 | 0.26 | 1.30 | 0.04 | 0.07 | 0.31 |
| **IGE** | 3.0 | 0.06 | 0.20 | 0.60 | 0.02 | 0.06 | 0.14 |



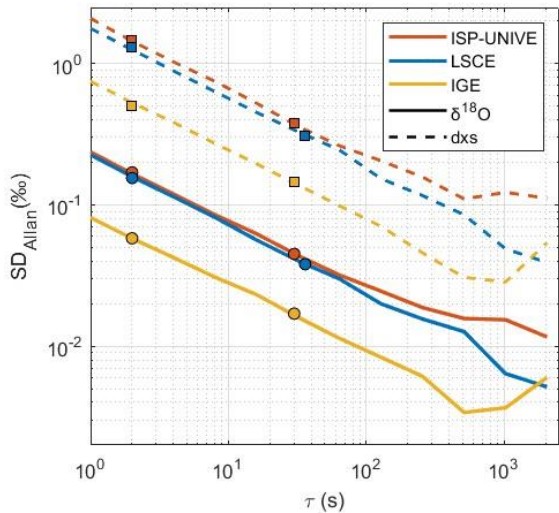

**Figure 4: Allan Deviation computed from 1-hour continuous UPW flow for ISP-UNIVE - orange (at 14.850±53 ppmv), LSCE - blue (at 20.380±190 ppmv) and IGE – yellow (at 18.100±105 ppmv) for $\delta^{18}O$ and d-excess. The dots and squares indicate the specific integration times ($\tau$), for each system, needed to melt 0.1 cm and 1.5 cm of the ice core at the nominal melting rate, respectively.**

### 2.4.3 Mixing in the CFA systems

The mixing in the CFA system attenuate the original precipitation signal, smoothing the isotope record (Gkinis et al., 2011). It occur at the melt-head due to water capillarity within the porous firn, in the debubbler, within the vaporiser chamber and the CRDS cavity, as well as throughout the transport tubing in both liquid and vapor phase (Jones et al., 2017a). To assess it, two impulsive responses are quantified using isotopic step functions. In practice, on the one hand, two ice sticks of different isotopic composition are melted in a row to derive the integrated mixing from the melt-head (MH) to evaluate $\sigma_{MH}$, and on the other hand, two isotopically distinct liquid samples are introduced at the selector valve (SV) level to determine $\sigma_{SV}$. The impulse response is determined by fitting a probability density function (PDF) over the first derivative of the Picarro response, described by the normal Gaussian:

$$f(x) = a_1 * exp\left(-\left(\frac{x - b_1}{c_1}\right)^2\right)$$

where $a_1$ is the amplitude, $b_1$ is the mean and $c_1$ is the standard deviation of the curve. Mixing lengths ($\sigma$) are defined as:

$$\sigma = \frac{c_1}{\sqrt{2}}$$

Additionally, the mixing in the liquid phase ($\sigma_L$) is calculated as the root square of the quadrature difference between $\sigma_{MH}$ and $\sigma_{SV}$ (Jones et al., 2017a).



## 2.5 Data Processing

The raw data from the Picarro analyser, recorded every second, are post-processed to align isotopic profiles with ice core depth. This includes converting the time scale to depth, calibrating isotopic values to the V-SMOW/SLAP scale, and filtering data affected by memory effects and artifacts. The building of the depth scale for the isotopic record is based on two computational steps common to all three laboratories. First, the timescale at MH is converted into a depth scale of the CFA sticks from the ice sticks' length measured before they are vertically loaded on the ice core-melting component, and from the continuous recording of the encoder position (located at the top of the ice sticks, which records at a frequency of approximately 1 Hz). Second, for each time step at the MH, the corresponding arrival time at the CRDS, conductivity cell, and additional fractionation collectors is calculated based on the flow rates of the peristaltic pumps. For the isotopic record, at ISP-UNIVE, similar to IGE, the delay from MH to CRDS signal recorded is computed through preliminary tests and subsequently corrected for pump rate changes. At LSCE, the arrival time for liquid phase is calculated using the continuously recorded flow rates of the two peristaltic pumps and the volumes associated with each component of the CFA setup. Additionally, the delay time for gas phase transport to the CRDS is estimated based on isotopic steps from the SV, under standard operating conditions (specifically, $N_2$ and water flow rates). This gas phase delay is assumed to remain constant throughout the duration of the CFA run. Conductivity profiles plotted on the common depth scale thus determined allow for validation of the process. For LSCE, which provides three conductivity profiles, validation is achieved through the effective superposition of conductivity features. At IGE, where a single device is used, validation relies on accurately aligning conductivity peaks matching the transitions between individual ice sticks with the logged depths. Lastly, readjustments may be needed between the actual stick length and the depth logged in the field, particularly for crumbling firn cores. Data at the beginning and end of CFA runs that are affected by mixing with pre- and post-circulated water are manually removed. For the ISP-UNIVE-CFA results presented in this study, data sections affected by humidity drops lower then ~8,000 ppmv were removed. These intervals were manually selected using a MATLAB graphical user interface, with short gaps (less than 5 minutes, ~ 15 cm depth) interpolated linearly possibly affecting the PSD analysis (see Sect. 3.3). For both ISP-UNIVE and IGE, the top 0.50 m of the first bag were removed due complications encountered during the initial melting phase, which included firn ice collapse, air bubbles and UPW intrusions resulting from the cleaning process of the tubing during line blockages. No additional data removal was performed in the data series presented for LSCE.

## 2.6 Comparison Procedure

The comparison between continuous and discrete isotopic records aims to highlight key technical differences in the CFA-CRDS setups and the operating procedures. Comparisons of profiles versus depth assess the agreement in calibration and depth scale attribution between laboratories, while the power spectral density (PSD) analysis – defined as the measure of signal's power content in the frequency domain - reveals system limitations caused by mixing and measurement noise. Before detailing the comparison in Sect. 3, we briefly describe the PSD approach (Fig. 5). Here, the discrete record is ideally considered as the




best approximation of the true signal preserved in the ice, limited only by discrete resolution and uncertainties in the depth for sampling cut. Since its spectrum lacks a flat region– typically indicative of white noise - at high frequencies, measurement noise never dominates over the signal. The discrete spectrum is flat at frequencies around 1 m⁻¹ (white area), dominated by precipitation intermittency and stratigraphic noise (Casado et al., 2020; Laepple et al., 2018). In contrast, it attenuates for scales between 0.5-0.9 m⁻¹ (yellow area), corresponding to diffusion (Johnsen et al., 2000). This firn diffusion, with length of 10-15 cm (Gkinis et al., 2021; Johnsen et al., 2000; Laepple et al., 2018; Whillans & Grootes, 1985), can be modelled by Eq. (1) (Johnsen et al., 2000):

$$P = P_o \, exp\left(-k^2 \sigma_{diff}^2\right)$$

where $\sigma_{diff}$ represents the firn diffusion length, and $k = 2\pi f$, with f being the frequency.

Mixing in the CFA behaves similarly to diffusion, albeit on a much smaller scale around 0.7-1.5 cm (Gkinis et al., 2011; Jones et al., 2017a). We theoretically simulate the continuous spectrum from the discrete one, by applying additional Gaussian smoothing to account for the CFA mixing length ($\sigma_{mix}$) and incorporating the measurement noise ($\varepsilon_N$) determined for the online analysis, as described by Eq. (2):

$$P = P_1 \, exp(-k^2 \sigma_{mix}^2) + \varepsilon_N$$

The simulated spectrum diverges from the discrete spectrum at the frequency where CFA mixing begins to affect the signal, showing smoothing in the medium frequency range (>0.5 m⁻¹, orange area) and flattering at higher frequencies due to measurement noise (brown area). Measurement noise generates a flat spectrum at the frequency where the Signal-to-Noise Ratio (SNR) equals 1 (Casado et al., 2020), permitting to determine the frequency limit where meaningful climatic information can still be retrieved as the point where the spectra of signal and noise intersect. Beyond this limit, noise dominates the signal, as the correlation between the record and the signal is defined as:

$$r^2 = \frac{SNR}{1 + SNR}$$

At SNR = 1, a minimum significant correlation $r = \sqrt{0.5} \sim 0.71$ is reached.



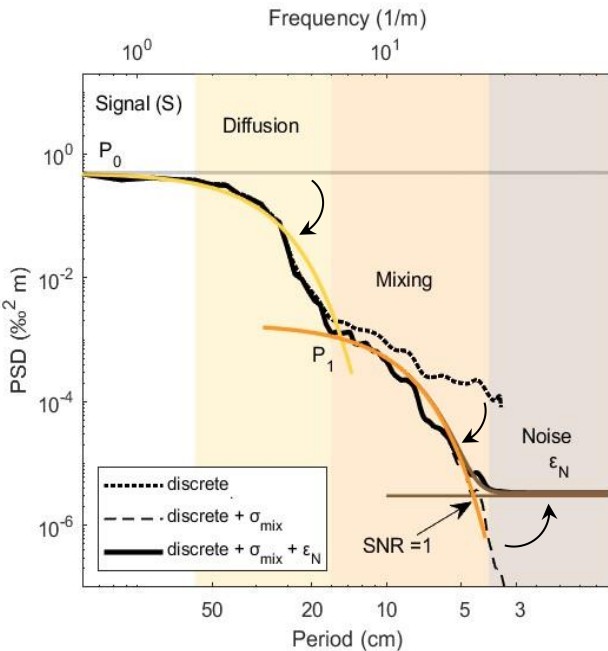

**Figure 5: Schematic PSD of discrete and simulated profiles. The simulated spectrum (solid black line) is obtained by combining the discrete power density (dashed black line) affected by natural diffusion in the firn ($\sigma_{diff}$ – yellow) with the mixing effect occurring in the CFA system ($\sigma_{mix}$ – orange) and the measurement noise associated with continuous measurements ($\varepsilon_N$ – brown).**

## 3 Results

In this section, we compare the measurements of the same 4-m section of the PALEO2 firn core by the CFA setups of ISP-
UNIVE, LSCE, and IGE with the discrete results. By analysing both in the depth and in spectral domains, we evaluate the precision and accuracy of each setup and operational procedure. In turn, we assess the effective resolution limit at which a reliable isotopic signal can be retrieved from the CFA outputs, based on each system's internal mixing and measurement noise.

### 3.1 Mixing Lengths

The quantified mixing lengths for the three inspected systems are based on step functions at level of the melt-head ($\sigma_{MH}$),
correspondent to the total mixing in the systems (defined as $\sigma_{mix}$ in the other Sections), and at level of the selector valve ($\sigma_{SV}$) as shown in Fig. 6 and Tab. 4. Previous studies used mock ice with varying isotope compositions to calculate $\sigma_{MH}$ (Dallmayr et al., 2024; Jones et al., 2017a). In this work, the PALEO2 firn core ($\rho = 0.58$ g cm$^{-3}$) has a lower density and higher porosity than the artificial ice ($\rho = 0.92$ g cm$^{-3}$), potentially leading to an underestimation of the mixing lengths when quantified based on ice-to-ice transitions. We compared the mixing lengths obtained from ice-to-firn, firn-to-ice, and ice-to-ice transitions
(Appendix B, Tab. B1), determining no significant differences in capillary action. Therefore, we used the mean value as the $\sigma_{mix}$ in our back-diffusion approach, and discuss the potential difference with the more realistic case of firn-to-firn transitions (see Discussion). The differences in isotopic composition between firn and mock UPW-ice are 40-50‰ for $\delta^{18}$O and 380-



400‰ for $\delta$D. The mixing lengths for $\delta^{18}$O and $\delta$D are very similar, so we present the results for $\delta^{18}$O. The mean $\sigma_{MH}$ values are 7.1 mm for LSCE, 11.3 mm for ISP-UNIVE, and 18.2 mm for IGE (Tab. 4). The mixing length in the liquid phase ($\sigma_L$) is

calculated as the root square of the difference in quadrature between $\sigma_{MH}$ and $\sigma_{SV}$. Values expressed in seconds are converted in millimeters, using the average melting rate. LSCE exhibits the smaller $\sigma_{MH}$ attributed to its shorter distribution line and the more efficient isotopic analysis setup. Conversely, IGE-CFA has the highest $\sigma_{MH}$ (Tab. B1) reflecting the higher volume debubbler and the longer distribution line. Notably, IGE exhibits a higher $\sigma_{SV}$ compared to LSCE, despite showing a lower Allan variance, which reflects the performance of the combined vaporiser-Picarro system.

The mixing lengths are consistent with the values reported by Jones et al. (2017a) at the Institute of Arctic and Alpine Research (INSTAAR) Stable Isotope Lab (SIL) and by Dallmayr et al. (2024) at the Alfred-Wegener-Institut Helmholtz-Zentrum für Polar-und Meeresforschung (AWI) for ice-to-ice transitions, equal to 7 mm and 13.6 mm, respectively (Tab. 4).

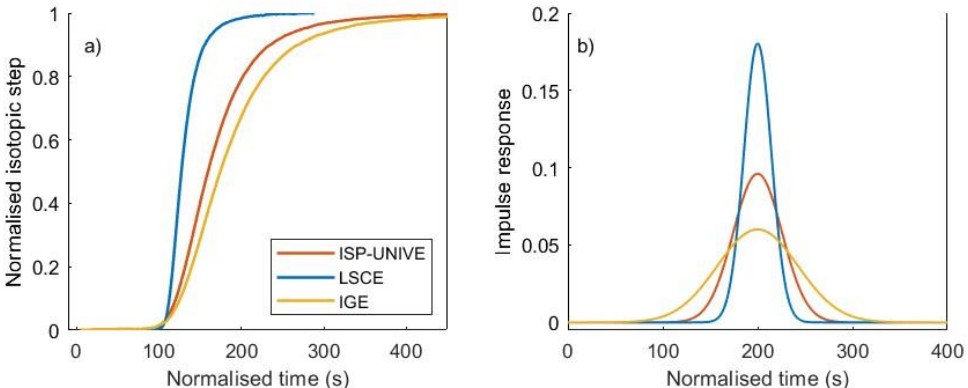

**Figure 6: a) The normalised transfer function and b) the normal PDF impulse response function for ice steps at level of the CFA**
**melt-head for $\delta^{18}$O.**

**Table 4: $\delta^{18}$O mixing lengths at melt-head level ($\sigma_{MH}$) and at selector-valve level ($\sigma_{SV}$) for the three different CFA-CRDS systems. The mixing length in the liquid phase ($\sigma_L$) is calculated as the difference in quadrature of $\sigma_{MH}$ and $\sigma_{SV}$. The mixing length expressed in seconds is converted in millimeters considering the average melting rate set at the three institutes. The 1SD are given in parenthesis. The mixing lengths are compared with Jones et al., 2017a at the Institute of Arctic and Alpine Research (INSTAAR)**
**Stable Isotope Lab (SIL) and Dallmayr et al., 2024 at the Alfred-Wegener-Institut Helmholtz-Zentrum für Polar-und Meeresforschung (AWI).**

|  | System | Melt rate (cm min⁻¹) | $\sigma_{MH}$ (s) | $\sigma_{MH}$ (mm) | $\sigma_{SV}$ (s) | $\sigma_L$ (s) | $\sigma_L$ (mm) |
|---|---|---|---|---|---|---|---|
| **This work** | ISP-UNIVE | 3 | 22.4 (2.0) | 11.2 (1.0) | 20.0 (0.6) | 10.1 | 5.1 |
|  | LSCE | 2.5 | 16.8 (2.2) | 7.1 (0.9) | 8.6 (1.4) | 14.4 | 6 |
|  | IGE | 3 | 36.4 (6.0) | 18.2 (3.0) | 16.8 (2.3) | 32.3 | 16.2 |
| **From literature** | INSTAAR | 2.5 | 17.4 (2.2) | 7 (0.9) | 9.4 (0.5) | 14.6 | 6 |
|  | AWI | 3.8 | 21.6 (2.4) | 13.6 (1.5) | 12.6 (1.8) |  | 4.5 |



## 3.2 Discrete vs Continuous Data

We present the continuous $\delta^{18}$O and d-excess records from ISP-UNIVE, LSCE and IGE in comparison with the discrete profiles
(Fig. 7). The CFA data, initially provided at 0.5 cm resolution after post-processing, have been integrated at the lower resolution that matches the discrete depth intervals to highlight the main differences (Fig. 7. a and c; the original data are presented in Appendix C, Fig. C1). The differences between the averaged continuous and discrete data are analysed using histograms of the differences at each depth point (Fig. 7. e and f). Overall, the variability in ice core $\delta^{18}$O records, primarily at the decimetric scale, is comparable between the three CFA profiles and the discrete sampling, showing no statistically
difference. The ISP-UNIVE-CFA results have no significant difference of -0.01±0.26‰ (mean ± 1σ) for $\delta^{18}$O. However, two sections - showing higher difference with discrete profile - were not removed during data post-processing. These intervals correspond to depths of 13.85-13.96 m and 15.62-15.85 m (Fig. 7a, orange areas). The first was caused by a humidity drop to 1,200 ppmv, where the isotopic record was removed and linearly interpolated. The second interval consisted in a humidity drop to 7,850 ppmv, slightly below the limit of water vapor mixing ratio imposed during typical analysis campaign, which was
retained to highlight the impact of humidity fluctuation. Although rare, such events occurred in the novel coupling of the CFA-CRDS at ISP-UNIVE, where achieving a highly consistent flow rate was challenging during initial analysis. For the LSCE-CFA, the mean difference for $\delta^{18}$O is slightly higher but remains non-significant, with a SD within the instrument's error (0.13±0.18‰ for $\delta^{18}$O, Tab. 5). The IGE-CFA results compared with discrete data show difference of -0.06±0.24‰ for $\delta^{18}$O. The SD, comparable to that of ISP-UNIVE but larger than that of LSCE, is primarily attributed to small shifts in the depth
scale. These centimeter-scale shifts are evident near the transitions between peaks (Fig. 7b). For d-excess, the ISP-UNIVE and IGE results show statistically difference with discrete data of -0.78±0.64‰ and 0.88±0.48‰, respectively. These discrepancies are mainly attributed to calibration, as supported by the reduced difference between discrete and CFA data after applying a calibration correction that shifts isotopic values to align the mean difference to zero (Fig. 7d). By contrast, the LSCE record show a non-significant difference of 0.03±0.55‰. Overall, the good agreement in both $\delta^{18}$O and d-excess between CFA and
discrete records indicates that the LSCE calibration process is the most robust among the three CFA setups for this series, which is why the entire PLAEO2 core was analysed at LSCE (Sect. 4).





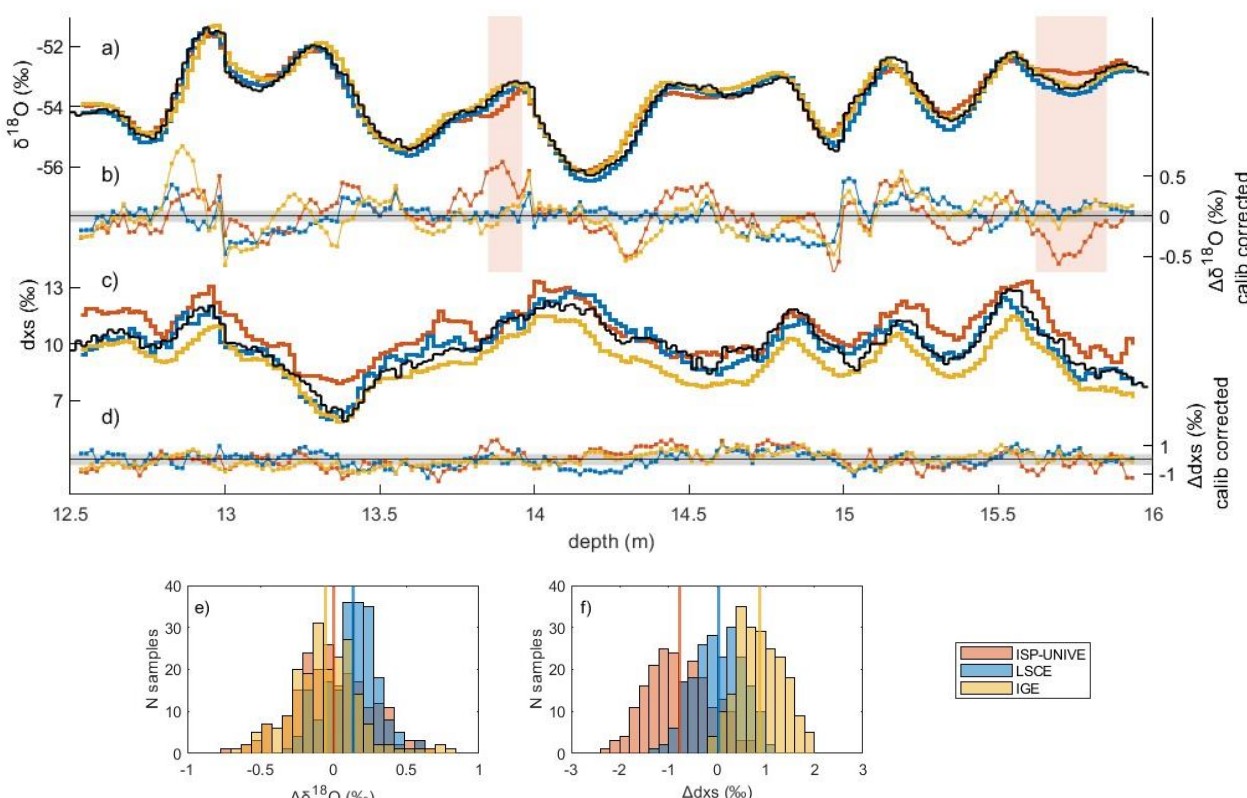

**Figure 7: a) and c) Comparison of discrete $\delta^{18}$O and d-excess profiles (black lines) for depth interval 12.5-16 m with integrated discrete record built from CFA measurements for ISP-UNIVE (orange), LSCE (blue) and IGE (yellow). The histograms e) and f) represent the distribution of difference between discrete data and discrete record built from CFA measurements. b) and d) show the difference between discrete data and discrete record built from CFA measurements corrected for calibration. The horizontal grey areas represent the uncertainties of the discrete analysis for $\delta^{18}$O and d-excess.**

**Table 5: Mean, SD and Root Mean Square (RMS) of the isotope difference between discrete and CFA data for ISP-UNIVE, LSCE and IGE. Significant differences (p < 0.05 using a Kruskall-Wallis nonparametric ANOVA test) are represented as bold.**

|  | $\Delta\delta^{18}$O (‰) | | | $\Delta$d$_{xs}$ (‰) | | |
|---|---|---|---|---|---|---|
|  | **Mean** | **SD** | **RMS** | **Mean** | **SD** | **RMS** |
| **ISP-UNIVE** | -0.01 | 0.26 | 0.26 | **-0.78** | 0.64 | 1.00 |
| **LSCE** | 0.13 | 0.18 | 0.22 | 0.03 | 0.55 | 0.55 |
| **IGE** | -0.06 | 0.24 | 0.25 | **0.88** | 0.48 | 1.00 |

## 3.3 Spectral Analysis

In this section, we conduct a PSD analysis on continuous and discrete results (Fig. 8) to assess the impact of mixing on continuous measurements and to determine the frequency limit at which the measurement noise begins to dominate over the



signal. Diffusion, with a length of 10-15 cm in firn, primarily smooths the climatic signal over period of 20-50 cm (yellow areas). Conversely, mixing within the CFA systems, characterised by lengths of a few centimeters, further attenuates the continuous signal over periods of 3-20 cm, as shown by the discrepancies between CFA and discrete spectra (orange areas). The frequency limit for reliable isotopic measurements corresponds to the intersection of the smoothed signal (black dashed lines S) and the noise flat spectrum (brown areas, black dashed lines $\varepsilon_N$), where the SNR equals to 1. These resolution limits are 1.8 cm for ISP-UNIVE, 1.6 cm for LSCE, and 1.3 cm for IGE. While the effects of mixing can be corrected by applying back-diffusion to the signal, attempting this on frequencies dominated by measurement noise would result in an artificial amplification of that noise. Back-diffusion applies an inverse Fourier transform, generated based on the nominal $\sigma_{mix}$ (Sect. 3.1), to the original record restoring the power of the higher frequencies. The back-diffused profiles show significant improvement in the amount of signal across the 3-10 cm period range for all three CFA systems. The lack of signal for periods ranging from 10 to 20 cm in the LSCE profile, and to some extent in the ISP-UNIVE, is not corrected by this back-diffusion, which does not act at such frequencies. Overall, as the signal is dominated by the lower frequency (1000 times more power at the 50 cm scale than at the 10 cm scale), this does not affect the performances of the comparison between the CFA setups and the discrete time series, as shown for the back-diffusion of IGE-CFA record (Appendix D, Fig. D1). This ensures that our approach accurately reflects the discrete signal and validates the effectiveness of the mixing length quantification through step function.

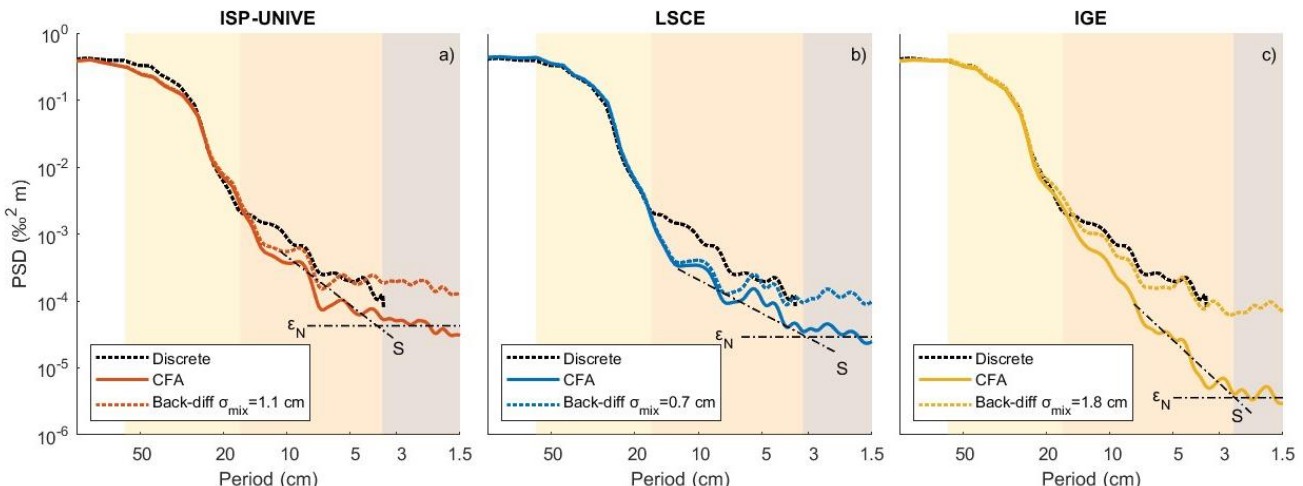

**Figure 8: PSD of continuous and back-diffused data compared to discrete spectrum for a) ISP-UNIVE, b) LSCE, and c) IGE. The crossing point of the dot lines, representative of the nearly flat white noise ($\varepsilon_N$) and the signal (S), corresponds at the SNR=1. For each system, the shading areas represent the period range at which the diffusion in firn (yellow), mixing in the CFA (orange) and measurement noise (brown) dominate.**



## 4 Discussion

The comparison of isotopic continuous and discrete records from the 4-m PALEO2 firn core highlights the advantages and limitations of the operating procedures and technical features of three CFA-CRDS systems developed at ISP-UNIVE, LSCE, and IGE (Tab. 6). We confirm that the primary advantage of the continuous method is its time efficiency. Using the LSCE-
CFA system, we were able to analyse the 18 m PALEO2 firn core in two days (Fig. 9). The cutting of the sticks was done in parallel to the melting, avoiding the storage of the firn for extended period of time and allowing for immediate resampling if needed. In contrast, discrete processing and analysis at 1.5 cm of the 4-m core section took over a month. Extrapolating this analytical payload, analysing the entire PALEO2 core would take more than five months. A notable limitation of the CFA setup is its application to the low-density upper part of the cores (specifically, the upper 1.67 m for the PALEO2 core), where
challenges in controlling melting speed and stick collapse require the use of discrete measurements rather than CFA. As previously found at other CFA equipped institutes, the precision for discrete and continuous flow analyses are equivalent (Emanuelsson et al., 2015; Gkinis et al., 2011; Jones et al., 2017a). Here, by using PSD to evaluate the effective resolution imposed by measurement noise, in addition to solely the precision, we were able to determine the resolution at which records can be interpreted: 1.8 cm, 1.6 cm and 1.3 cm for ISP-UNIVE, LSCE and IGE, respectively. The primary factors affecting the
precision and the resolution are the performance of the Picarro instrument and the humidity level maintained during the analysis. Maintaining a constant high humidity level is crucial for mitigating performance variations (Davidge et al., 2022), as this was identified as the key factor responsible for the difference between the ISP-UNIVE-CFA and discrete results. Additionally, the CFA method can be limited by mixing within the system, requiring signal integration over several millimeters of depth to accurately reflect the isotopic composition of the ice.
We validated the mixing lengths evaluated by step function through PSD analysis of the back-diffused profile for each of the three systems, confirming that the $\sigma_{mix}$ derived from ice-to-firn, firn-to-ice, and ice-to-ice transitions serves as a reliable surrogate for the realistic case of firn-to-firn transitions. Furthermore, although our study focused on low-density firn cores, we expect these findings to remain valid - or potentially improve - for deeper core sections, given the reduced mixing that is likely to occur at the melt-head level. The IGE-CFA system, characterised by longer distribution line, a 1,000 µl volume
debubbler, and two dust filters, results in the longer $\sigma_{MH}$ (1.8±0.3 cm). However, this increased mixing length is compensated by the low noise level of the high-performance analyser, which enhances effective resolution.

Back-diffusion can effectively mitigate the mixing effect at high frequencies. However, while the corrected data show a ~0.1‰ gain in isotopic signal in proximity of the climatic peaks, the improvement compared to the discrete profile is very limited. This is consistent with the PSD of both continuous and discrete records, where the signal is dominated by longer periods (20–
100 cm, with a power of $10^{-1}$-$10^{0}$ ‰$^2$ m); while at mixing scale (10–20 cm) it is associated with lower power ($10^{-3}$-$10^{-4}$ ‰$^2$ m) comparable to the instrumental limit. These findings suggest that the impact of back-diffusion correction is limited for cores recording inter-annual to decadal signals (Casado et al., 2020; Münch & Laepple, 2018). However, it may become crucial in high accumulation areas for records with greater variability at shorter depth scales, such as deeper core sections where decadal




signals are compressed into smaller depths. It is also relevant when analysing significant events, like those associated with
atmospheric rivers (Wille et al., 2022), in areas where firn diffusion does not significantly erase the signal, unlike at the PALEO
site. Furthermore, removing the effects of firn diffusion occurring prior to ice core drilling will require the application of back-
diffusion function, along with accurate estimates of variability on centimetric scales (Jones et al., 2017b).

The LSCE-CFA, characterised by the lower discrepancy compared to discrete results (Fig. 7 e and f), has a maximum potential
resolution of 0.7 cm, according to the mixing length. However, the actual achievable resolution for this specific measurement
campaign is limited to 1.6 cm due to the relatively low performance of the vaporiser-Picarro system. This suggests that using
a better Picarro analyser, characterised in particular by a better Allan variance, could significantly improve the LSCE-CFA
setup's performance. Here, we argue that the signal measured with the LSCE-CFA from the PALEO2 firn core (Fig. 9)
accurately reflects the isotopic variability present in the firn, both in terms of $\delta^{18}O$ and d-excess. Furthermore, the results
obtained can be confidently compared to cores measured at the other institutes.

An additional limitation of CFA systems is data loss at the beginning and end of the analysis due to the memory effect of the
CRDS analyser. We strongly recommend using an ice stick with an isotopic composition similar to that of the samples before
and after the analysis rather than mock UPW ice, as recently suggested by Davidge et al. (2022). For firn analysis, filling the
tubes with depleted water of similar isotopic composition to that of the core helps mitigate transitions at the MH during the
switch from ice to firn. Since memory effects decay exponentially, the length of data that must be removed to eliminate this
effect depends on the isotopic composition difference. At IGE, a mock UPW ice with $\delta^{18}O$ composition of -12‰ precedes ice
core analyses. For a $\delta^{18}O$ difference of about 40‰ between mock and sample ice, approximately 9.5 cm of the record must be
removed. Conversely, for an ice stick with $\delta^{18}O$ difference of 3-4‰, only the first 1.8 cm are affected by memory effect.

To ensure reliable isotopic records, we list recommendations for future improvement or setting up of a CFA-CRDS system:

- First, developing a short transport line and then selecting a low-volume debubbler can reduce mixing.
- Installing conductivity devices both downstream of the MH and upstream of the CRDS instrument ensures accurate
depth-scale attribution.
- Using ice sticks or filling the tubes with water of isotopic composition similar to the samples, rather than UPW ice,
helps reduce memory effects. In addition, avoiding the use of UPW water in the system during the analysis for clean
the line minimises data loss.
- Conduct pre-analysis testing of the core density to determine the appropriate MH temperature for achieving the
desired flow rate and select a melting rate that allows a long integration time to enhance measurement precision.
- Maintain constant high humidity levels for isotope analysis (optimal range for the Picarro instrument: 18,000-22,000
ppmv) mitigates performance variations (Davidge et al., 2022; Gkinis et al., 2010).

**Table 6: Summary of the main advantages and limitations identified in the CFA-CRDS comparison.**

| | Values | Comments |
|---|---|---|
| **Advantages** | | |



| Time efficiency | 1.5-1.8 m h$^{-1}$ | CFA reduces the time needed to analyse deeper ice cores. |
|---|---|---|
| Precision | 0.02-0.05‰ for $\delta^{18}$O, 0.06-0.16‰ for $\delta$D | The precision is primarily related to Picarro performance and secondarily by the humidity level maintained during analysis. The optimum level for Picarro is 18,000-22,000 ppmv. |
| High-resolution | 1.3-1.8 cm | The limit resolution is primarily determined by the measurement noise. |
| **Limits** | | |
| Mixing | 0.7-2 cm | The mixing effect can be restored at a period of about 5 cm. The error introduce is 0.06‰ for $\delta^{18}$O. |
| Data gaps | Approx. 1 to 15 cm length | At the beginning and end of each day's analysis, in the event of air bubble intrusion during the analysis or related to melting problems. Gaps of a few cm are filled by interpolation or additional discrete resampling of the core. |


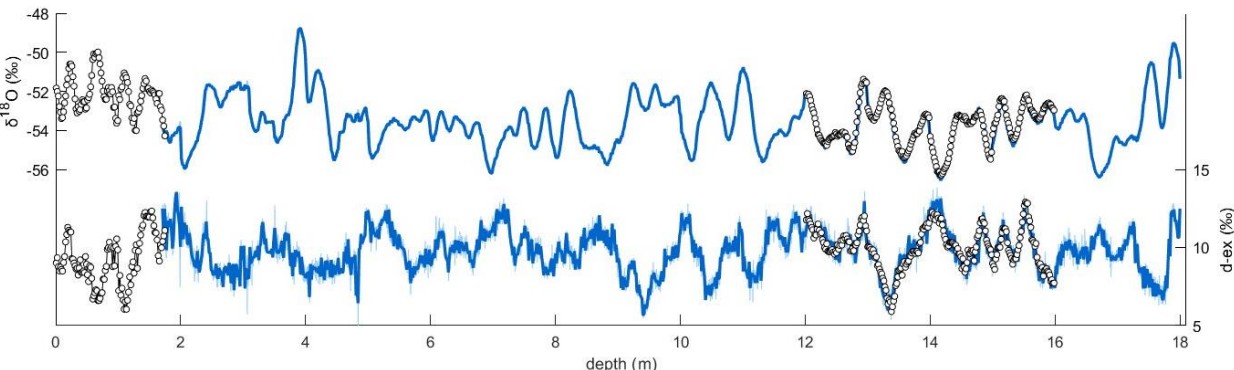

**Figure 9: The 18 m PALEO2 core analysed with CFA-CRDS system at LSCE. The $\delta^{18}$O and d-excess profiles of the raw data (light blue) and block-averaged at 1.6 cm data (blue) are shown with the discrete results (white dots).**

## 5 Conclusions

The technical intercomparison of the three CFA-CRDS systems developed at the ISP-UNIVE, LSCE and IGE, highlighted
each system's strengths and identified opportunities for future optimisation and development. Operating at a melt rate of 2.5-3 cm min$^{-1}$ under stable conditions, these systems measure 8-10 m of firn core per day, confirming that CFA-CRDS is a fast, high-resolution method for isotope analysis, with precision comparable to the discrete method. For ISP-UNIVE-CFA and LSCE-CFA systems, the main limitation in achieving higher resolution is imposed by the performance of the analyser. Conversely, the IGE-CFA system, with the great advantage of providing chemistry measurements online, has the setup
limitation related to its longer distribution line and higher volume debubbler, which promotes mixing. To optimise system performance, it is crucial to set a melt rate high enough to ensure sufficient flow for all instruments and fraction collectors in the CFA system, while minimising excessive mixing at the melt-head and in the tubing, and without extending the duration of the CFA campaign for melting the core. At the same time, the melt rate must not be too high, as this would require shorter integration times - thereby compromising precision - and resulting in mixing affecting longer depth intervals, when it is
converted from seconds to centimeters.



Finally, the outcomes of this work gave the basis for the post-processing of four PALEO cores analysed at ISP-UNIVE, LSCE, and IGE as part of the EAIIST (Traversa et al., 2023; Ventisette et al., 2023), allowing the climate signal to be interpreted by reconstructing it at the highest retrievable resolution.

## Appendix A: CFA-CRDS systems

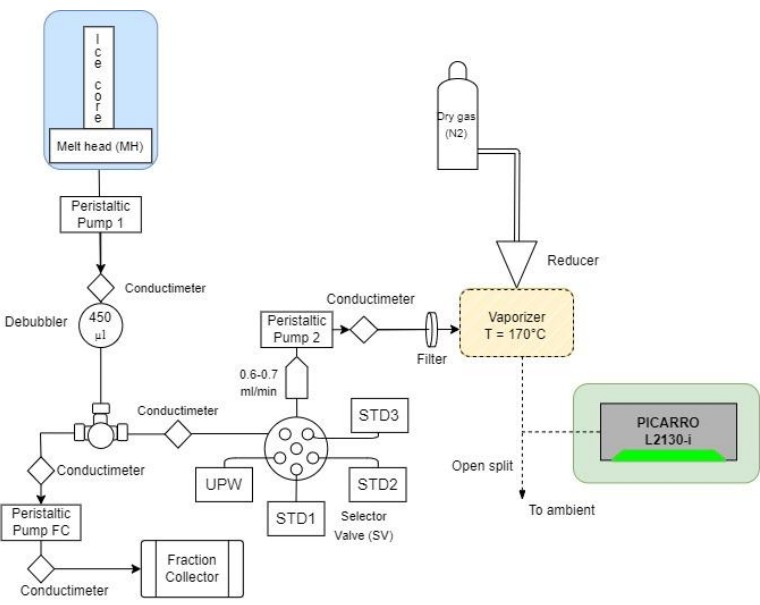


**Figure A1: Schematic of the CFA coupled with CRDS at LSCE.**

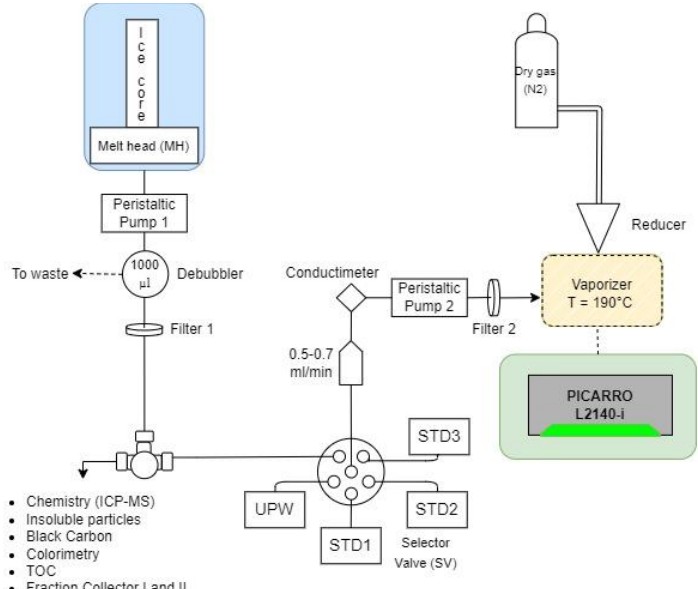




**Figure A2: Schematic of the CFA coupled with CRDS at IGE.**

**Appendix B: mixing lengths evaluated at MH and SV levels**

**Table B1: $\delta^{18}$O mean mixing lengths ($\sigma_{MH}$) at level of the melt-head derived from the normal PDF calculated for N number of firn-to-ice, ice-to-firn and ice-to-ice isotopic steps. The $\sigma_{MH}$ is the mean value of all the steps for each institute. The lengths are expressed in mm accounting for the nominal melting speed and the respective 1SD are in parenthesis. The $\sigma_{SV}$ is obtain from the steps at selector valve level.**

|  | $\sigma_{MH}$ firn-ice (mm) |  | $\sigma_{MH}$ ice-firn (mm) |  | $\sigma_{MH}$ ice-ice (mm) |  | $\sigma_{MH}$ (mm) | $\sigma_{SV}$ (s) |  |
|---|---|---|---|---|---|---|---|---|---|
| **ISP-UNIVE** | N=4 | 10.5 (0.6) | N=3 | 12.2 (0.3) |  | - | 11.2 (1.0) | N=4 | 20.0 (0.6) |
| **LSCE** | N=2 | 8.4 (0.1) |  | - | N=5 | 6.4 (0.5) | 7.1 (0.1) | N=4 | 8.6 (1.4) |
| **IGE** |  | - | N=6 | 18.2 (3.0) |  | - | 18.2 (3.0) | N=8 | 16.8 (2.3) |

**Appendix C: CRDS processed data at 0.5 cm in comparison with record constructed at 1.5 cm and discrete data**

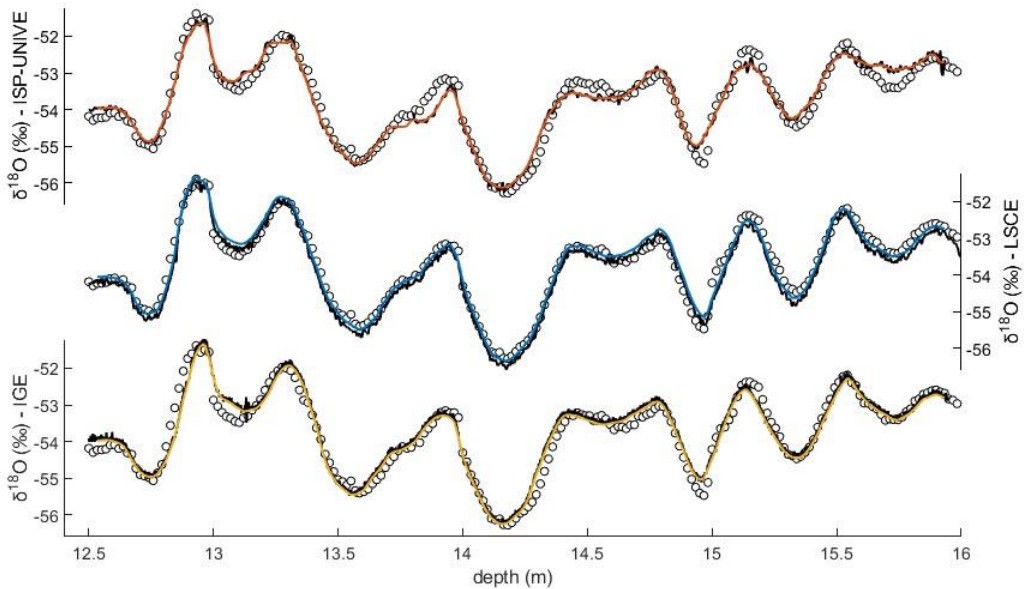

**Figure C1: Comparison of $\delta^{18}$O from CFA post-processed data at 0.5 cm resolution (black lines) and discrete values (white dots). Colored lines show discrete records constructed from CFA measurements at a 1.5 cm resolution, after calibration correction.**

**Appendix D: Sensitivity test of the evaluation of the mixing length**

Back-diffusion can mitigate the mixing effect at higher frequencies. To quantify the gain in isotopic composition, we present the continuous $\delta^{18}$O IGE profile back-diffused with $\sigma_{mix}$ = 1.8±0.3 cm (Fig. D1). To prevent amplification of measurement





noise, high frequencies with a signal-to-noise ratio (SNR) below 1 are removed by block averaging the profile at the specified frequency limit. Block averaging aggregates isotope records onto a new depth scale resolution, defined by each system's frequency limit. The original data are averaged within each interval and assigned to the corresponding depth point

on the new scale. When comparing the raw data and the back-diffused profiles, we observe a gain of up to 0.1‰ in proximity of the climatic peaks. However, since trough PSD analysis both continuous and discrete records show very low power at mixing length scale compared to the power of the dominant signal, it is evident that the back-diffusion approach has a limited impact on restoring the signal, when compared to the discrete profile.

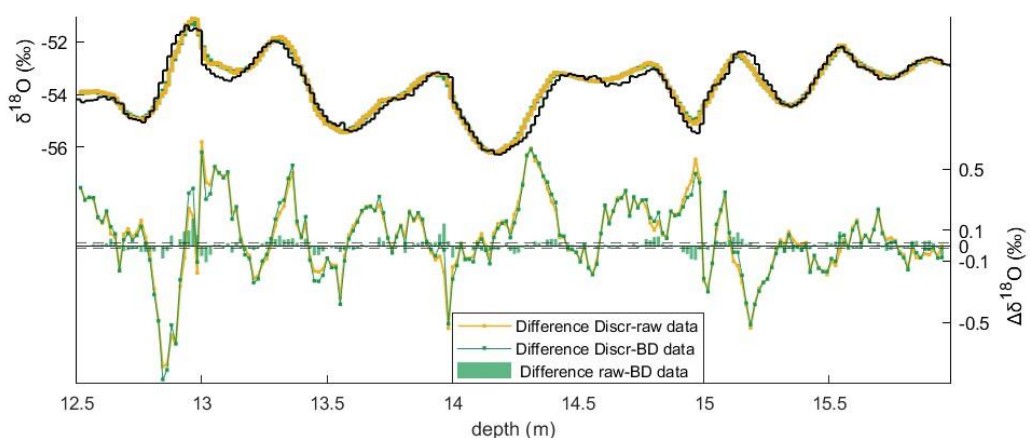


**Figure D1: (Top) Discrete $\delta^{18}$O record (black) compared with IGE-CFA profile (yellow) and back-diffused profile using $\sigma_{mix}$= 1.8 cm (green). (Bottom) The difference between discrete record and the original CFA profile (yellow line) and the back-diffused profile (green line). Green bars represent the difference between raw data and back-diffused data. The horizontal dashed lines represent the IGE instrument SD (±0.02 ‰) for integration time of 30 seconds (required to melt 1.5 cm of ice).**

**Author contributions**

AP, AS, DZ and JG developed the ISP-UNIVE-CFA-CRDS and performed the measurement in Venice; EF, RJ, OJ, AL developed the LSCE-CFA-CRDS system; AP, EF, RJ, TT and MC performed the measurement in Paris; EG, PDA, FP, JS and PG developed the IGE-CFA-CRDS and performed the measurement in Grenoble; AP and MC analysed the data and wrote the manuscript draft; EF, PDA, AL, AS, EG, CM and BS reviewed and edited the manuscript.

**Competing interests**

The authors declare that they have no conflict of interest.



**Acknowledgements**

We thank the research groups at ISP-UNIVE, LSCE and IGE for their technical support and the resources made available for
the analyses in their laboratories. We acknowledge the ANR EAIIST project, grant ANR—16-CE01-0011-01 of the French
Agence Nationale de la Recherche, the BNP-Paribas foundation and its Climate Initiative Program, the Institut Polaire Français
IPEV, the MIUR (Ministry of Education, University and Research) - PNRA (National Antarctic Research Program) through
the "EAIIST" PNRA16_00049-B and "EAIIST-phase2" PNRA19_00093 projects. We thank the MPM logistic team from
Seriate for the special transportation of the processed ice samples.

**Financial support**

This work was supported by the Polar Science PhD scholarship and Erasmus+ program.

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
