# Peer review of "Interlaboratory Comparison of Continuous Flow Analysis (CFA) Systems for High-Resolution Water Isotope Measurements in Ice Cores"

_EGUsphere, 2024_

## Referee Comment (RC2)

**Review: An International Intercomparison of Continuous Flow Analysis (CFA) Systems for High-Resolution Water Isotope Measurements in Ice Cores by Petteni et al.**

**I. Overview**

The manuscript *"An International Intercomparison of Continuous Flow Analysis (CFA) Systems for High-Resolution Water Isotope Measurements in Ice Cores"* by Petteni et al. deals with the comparison of three Continuous Flow Analysis systems developed for measuring the water isotopic composition ($\delta^{18}$O, $\delta$D) of ice core samples. The study focuses on comparison tests, providing quantitative evaluations of the resolution and precision achieved by the three systems. Example data from a firn core from Antarctica are used to support some of these tests.

This is a relevant contribution and it fits within the scope of the AMT journal. The manuscript is of good quality in terms of its methods and presentation; however, it lacks clarity and presents some important flaws and misconceptions, especially with respect to the diffusion and deconvolution parts. Therefore, I recommend publication after significant and major revisions are considered.

**II. General remarks**

I think it is important for the authors to state early in the paper that this is an intercomparison using firn cores. While it is understandable that these systems can be used for ice cores, a significant part of the paper deals with sample diffusion effects, for which the porosity of the core sample is of immense importance. The term "international" should also be reconsidered and replaced with something more appropriate.

The flow of the paper is not entirely smooth, and there are sections from the Methods and Results that could, in principle, be bundled together. For example, Sections 2.6, 3.1, and 3.3 all deal with sample diffusion, signal attenuation, and spectral methods. The titles in their current form are vague and somewhat misleading.

Some of the nomenclature used is atypical or incorrect. The authors use the term "mixing" extensively. This should be replaced with diffusion/dispersion/signal attenuation, as "mixing" describes a very different process governed by different models and mathematical frameworks. The same applies for the term The term "integrated" when referring to the averaging of the data over larger intervals.

An important aspect I struggled with while reading the paper was the sampling resolution of the datasets and the corresponding Nyquist frequency in the plots presenting data in the spectral domain. Please state clearly what your $\Delta x$ is, and clarify the 1-s sample acquisition time of the CRDS instruments. Typically, Picarro spectrometers export data at non-fixed time intervals. In all plots presenting spectral data, ensure that the Nyquist frequency/period is clearly indicated. Currently, the PSDs of the discrete data, for example, do not extend to 33.3 $\text{m}^{-1}$. What is the explanation for this?

Lastly, some of the central points of the paper regarding noise estimation and sample dispersion are based on calculations and data that are not clearly described and lack mathematical clarity. For example, the authors only present the fits to the impulse responses in Fig. 6 without specifying what exactly the reader is seeing. The Allan variance calculation also lacks explanation, and based on the shape and smoothness of the lines in Figure 4, it is quite clear that it has not been done correctly.

Many of the claims regarding the mathematical treatment and the calculation of the transfer functions involved are incorrect, particularly in Sections 2.5 and 3.3. The manuscript lacks a clearer mathematical foundation to support these claims. The same applies to the comparison between the CFA time series and the discrete data. The evaluation of the results is largely subjective.

**III. MORE SPECIFIC ON THE VARIOUS SECTIONS**

**A. Experimental**

The manuscript can benefit from more clarity and depth relating to the desciption of the experimental part. This is very important for a submission to AMT. There is absolutely no information about the three vaporizers, a critical part of the system with respect to precision and achievable resolution. Are they all based on the capillary method? What is the diameter and the length of the capillary. What is the bore diameter of the tee split in the vaporizer? Also information about the specifics of other parts of the system like the filter used. I would appreciate all the diagrams to be moved in the same section in the main text.

Since this paper assesses the performance of the systems based on firn analysis a subsection dedicated to the melters is essential. Capillary effect in the firn can signifficantly affect the dispersion of the sample and the choice of the melter design can have a big impact on these effects. Please provide the drawings of the three different melters. The same should apply for the design of the debubblers.

**B. On water concentration and Allan variance**

In Section 2.4.1, the authors conflate two distinct effects of water concentration on isotope spectroscopy. The first relates to the choice of water vapor concentration that yields optimal precision. The second, as described in [2], concerns the dependence of the water isotope ratio signal on water concentration, which required a linear correction in that study. The selection of the 15,000–22,000 ppmv range in [2] pertains solely to the linearity of this dependence and the need for correction. However, the instrument examined in [2] is older and fundamentally different from the Picarro variants used in the present study.

The authors choose to show results only from the Venice system. As this is a technical intercomparison study, I believe results should be shown for all three systems and for both $\delta^{18}O$ and $\delta D$.

The Allan variance analysis appears problematic. Given the acquisition rate of the Picarro 2130 and 2140 models (2 Hz), the Allan variance curves should exhibit more high-frequency structure than the very smooth lines presented in Fig. 4. It seems clear that something else is being computed. Please consult [2, 4, 7, 8] for relevant plots and formulas. Additionally, a comment on how the authors transition from the non-fixed acquisition rate of the Picarros to a fixed timestep would be appreciated. Are the Picarros "pinged" at a constant interval via external control software, or is the data interpolated post-acquisition? The manuscript mentions that the results in Fig. 4 are based on at least 2-hour injections of UPW. Why do the Allan variance curves stop at approximately 2000 seconds and not extend to at least 3600 seconds, which would be the expected upper limit for $\tau = t_{acq}/2$ [1]?

It would be helpful to see the code used for this calculation, or at least a clear mathematical formulation. The $\delta^{18}$O and $\delta$Dtime series used in the Allan variance calculation should also be shown.

For a technical intercomparison paper of this nature, one would expect a deeper analysis of the mechanisms underlying the significantly better precision observed in the Grenoble system. Water concentration level is not a plausible explanation, as it is comparable to that of the LSCE system. To state that "the IGE system exhibits higher precision, attributed to better instrument performance as indicated by Allan deviation" is a rather cyclical argument.

*C. Sample diffusion*

The diffusion of the sample in CFA systems and the resulting attenuation of the signal power is an important artefact that must be addressed. There are various approaches to this issue, one of which uses spectral methods and transfer functions estimated from impulse responses and/or step functions. The manuscript presents some of these aspects in Sections 2.4.3, 2.6 (whose title should certainly be reconsidered), 3.1, and 3.3. First, I find the term "mixing" misleading, as it technically refers to the blending of different compounds. Therefore, terms like diffusion, dispersion, and signal attenuation should be used throughout the manuscript.

In Section 2.4.3, the manuscript describes how $\sigma_L$ can be calculated, referring to [6]. However, based on the schematics of the three systems, $\sigma_{sv}$ is not equivalent to vapor diffusion as defined in [6]. Downstream of the selection valves, there are peristaltic pumps and filters in both the LSCE and Grenoble systems, all contributing to liquid-phase dispersion. The reason why $\sigma_{sv}$ is equivalent to $\sigma_{vapor}$ in [6] is that, in that setup, the tubing downstream of the selection valve is minimal and leads directly to a vaporization unit (nebulizer), eliminating the need for a pump. How does this significant detail affect the calculation of sample diffusion in the present study?

Further on, in Section 2.6, the manuscript describes a modelling approach for the spectrum of the CFA data. The approach is problematic, as it assumes that CFA-induced diffusion adds power to the signal, represented by the term $P_1$. This is not physically possible. Diffusion does not add power—it only removes it.

Another important aspect missing from the analysis is the diffusion induced by discrete sampling itself. A sampling interval of 1.5 cm is roughly equivalent to a Gaussian transfer function with a diffusion length of 0.5 cm [5].

Throughout the manuscript, there is no information provided on the ice core site characteristics like temperature, accumulation and surface density. How do the authors estimate a firn diffusion length of 10–15 cm? Is this ice-equivalent, or does it refer to firn density at the sampled depth?

In Section 3.1, the reader is presented with step functions and impulse responses, but without access to the underlying data or fits. For a technical publication like this, Fig. 6 should incorporate those elements. The information that the authors have used a sequence of firn–ice samples in various combinations is important. Expected differences in diffusion characteristics due to capillary effects and firn porosity should be discussed.

One of the most interesting results of the study—but insufficiently investigated—is why the LSCE system shows more diffusion downstream of the selection valve compared to the segment from the melter to the selection valve (8.6 s vs. 14.4 s, Table 4). The other two systems—and every system I am aware of—show the opposite behavior. Additionally, the LSCE system does not appear to be fundamentally different from the others. This is something the authors should look into.

Regarding Section 3.3 (Spectral analysis), I have several comments. First, it lacks a clear description of the mathematical foundation. The text describes the deconvolution step as an inverse Fourier transform, but it is

not specified what exactly is being transformed. Do the authors construct a restoration filter? Is it optimized for measurement noise as in [3]? The text lacks both mathematical clarity and detail.

There are also misconceptions regarding the influence of the various transfer functions (firn/CFA) on signal attenuation. A transfer function with a diffusion length of 15 cm has a much greater impact (several orders of magnitude) on cycles with periods of 3–20 cm (5–33 $\mathrm{m}^{-1}$) than the CFA transfer function with a diffusion length of 1.5 cm. See plot below. So why do the authors claim that diffusion with a diffusion length of 10-15 cm primarily smooths the climatic signal over periods of 20-50 cm?

The authors claim a significant improvement in the 3–10 cm cycle range due to back diffusion correction, but no data are shown to support this. The data shown in Appendix D indicate the effect is negligible. Which is it?

How is it possible that the measured signals lack cycles in the 10–20 cm range? A quick inspection of both the measured profiles and their power spectral densities reveals significant power in these frequencies. At the same time, a Wiener restoration filter for deconvolving ice core data with diffusion lengths of 13.4 and 16.4 mm is shown in Fig. 2 of [3]. It is clear that both these back diffusion filters—with values very similar to those in the current study—act extensively in this frequency range. Can the authors elaborate?

Clarifying these questions requires presenting the mathematics used—how is the restoration filter constructed, and what does it look like in the frequency domain?

**D. Discrete vs Continuous**

In the comparison between the produced time series, the terms "statistical difference" and "significant difference" are used. I believe it is important that the authors explain these terms and clarify what objective test they use for statistical significance. A sound normality test for the residuals between all the time series would greatly improve the manuscript. The Shapiro-Wilk and Anderson–Darling tests are some possible choices.

I believe that the manuscript needs extensive work in the review phase addressing these key points, therefore I will not add more minor comments in this review.

**REFERENCES**

[1] F. Czerwinski, A. C. Richardson, and L. B. Oddershede. Quantifying noise in optical tweezers by allan variance. *Optics Express*, 17(15):13255–13269–13255–13269, 2009.

[2] V. Gkinis, T. J. Popp, S. J. Johnsen, and T. Blunier. A continuous stream flash evaporator for the calibration of an IR cavity ring-down spectrometer for the isotopic analysis of water. *Isotopes In Environmental and Health Studies*, 46(4):463–475, 2010. doi: https://doi.org/10.1080/10256016.2010.538052.

[3] V. Gkinis, T. J. Popp, T. Blunier, M. Bigler, S. Schupbach, E. Kettner, and S. J. Johnsen. Water isotopic ratios from a continuously melted ice core sample. *Atmos. Meas. Tech.*, 4(11):2531–2542, 2011. doi: https://doi.org/10.5194/amt-4-2531-2011.

[4] V. Gkinis, S. Jackson, N. J. Abram, M. Curran, T. Blunier, M. Halan, H. A. Kjær, A. Moy, K. M. Peensoo, T. Quistgaard, T. R. Vance, and P. Vallelonga. An 1135 year very high-resolution water isotope record of polar precipitation from the Indo-Pacific sector of East Antarctica. *Australian Antarctic Data Centre*, 2024. doi: http://dx.doi.org/doi:10.26179/ygeq-1a95.

[5] Christian Holme, Vasileios Gkinis, and Bo M. Vinther. Molecular diffusion of stable water isotopes in polar firn as a proxy for past temperatures. *Geochim. Cos-*

Fig. 1. Diffusion transfer functions

[Figure]

*mochim. Acta*, 225:128–145, 2018. doi: https://doi.org/10.1016/j.gca.2018.01.015. URL http://www.sciencedirect.com/science/article/pii/S0016703718300188.

[6] T. R. Jones, J. W. C. White, E. J. Steig, B. H. Vaughn, V. Morris, V. Gkinis, B. R. Markle, and S. W. Schoenemann. Improved methodologies for continuous-flow analysis of stable water isotopes in ice cores. *Atmos. Meas. Tech.*, 10(2):617–632, 2017. doi: https://doi.org/10.5194/amt-10-617-2017.

[7] E. J. Steig, V. Gkinis, A. J. Schauer, S. W. Schoenemann, K. Samek, J. Hoffnagle, K. J. Dennis, and S. M. Tan. Calibrated high-precision [17]O-excess measurements using cavity ring-down spectroscopy with laser-current-tuned cavity resonance. *Atmos. Meas. Tech.*, 7(8):2421–2435, 2014. URL http://www.atmos-meas-tech.net/7/2421/2014/.

[8] P. Werle. Accuracy and precision of laser spectrometers for trace gas sensing in the presence of optical fringes and atmospheric turbulence. *Applied Physics B-lasers and Optics*, 102(2):313–329–313–329, 2011. doi: 10.1007/s00340-010-4165-9.

Fig. 2.  Restoration filters

[Figure]

---

## Author Comment (AC1)

Dear editor,

We would like to thank the two reviewers for their comments. We have been working toward a new version of the manuscript taking their respective comments into account. We include and number the reviewers' comments in **black**. The comments from Reviewer 1 are numbered from 1 to 14, and those from Reviewer 2 from 15 to 36. The referees' comments have been addressed individually, as requested by the journal. When both reviewers address the same topic, we indicate the corresponding reference number(s).
Our responses are in blue, and the modifications to the manuscript in red in this response file.

On behalf of all the co-authors,
Agnese Petteni

Review of "An International Intercomparison of Continuous Flow Analysis (CFA) Systems for High Resolution Water Isotope Measurements in Ice Cores" by Agnese Petteni et al.

**General comments:**
The submitted manuscript presents an intercomparison of coupled CFA-CRDS systems for high resolution water isotope analysis in ice cores, involving two French laboratories (LSCE, IGE) and one Italian laboratory (ISP-UNIVE). The CFA is now an established technique to analyze ice cores, but has inherent mixing limits. Thus, the use of the power spectral density (PSD) approach to quantify the analysis resolution is of strong interest for the climate reconstruction based on ice cores.
Moreover, the study provides practical recommendations to optimize the CFA-CRDS setups, making this work relevant for colleagues in ice core analysis. I would like to see this work published, but only after taking care of the following revisions:

Several parts of the manuscript shall be re-written
• Paragraphs are sometimes pretty heavy to read, due to too long sentences
• I had the feeling that some messages are repeated
• Information are mixed within sections (Intro, Method, Results).
The manuscript has been partly re-written, refer to the specific comments for more details.

1) Despite the (surprising) small differences with the other systems' results, the very low humidity level of the ISP-UNIVE CFA may most likely be source of the (1) increased measurement noise shown by the Allan deviation test, and especially (2) a bias in the measurement as water vapor mixing ratios below <15000ppm leads to non-linearity in the instrument response (Gkinis, 2010). I agree that the instruments used in this study give better performances than the one used back then, which likely explains why the results are that good. Nevertheless, Fig 3 clearly shows a significant trend towards higher value (especially for dD and thus also dexcess), and I would be very surprised if the measurement stays the same at 20000ppm… Thus, to me the conclusions using these data are questionable in the current state.

We agree that the ideal measurement range for the Picarro instrument is around 20,000 ppmv, rather than 10 to 14,000 ppmv conditions. We were obliged to work at this level at ISP-UNIVE, as the water flux provided at the vaporizer did not allow us to reach higher constant humidity levels. And indeed, since modern Picarro instruments do not show strong non-linearity in the instrument response down to 10,000 ppmv now (for instance Aemisegger et al, 2012 compared a L1115i to a L2130i and really showcase the improvement compared to 2010), a big part of the bias is removed, but still, the precision of the instrument is going to be lower at the humidity levels we studied compared to 20,000 ppmv. In the updated manuscript, we will provide Allan variance plots at different humidity levels to illustrate which part of the lower performances of the Venice setup can be explained by the lower humidity.
The new figure will include similar data to the figure below, but for longer time intervals:

[Figure]

Fig. R1. d18O and dD continuous analysed (top) at different humidity levels (bottom)

[Figure]

Fig. 3. Mean δ18O and δD values for the 3-minute intervals selected at humidity level of 8,000, 9,500, 11,500 and 14,000 ppmv in the ISP-UNIVE setup (Fig. R1 - grey intervals). The confidence levels are defined as the Allan Deviation for integration time of 1 s (±1σ).

This will also allow us to assess the potential trend toward 20,000 ppmv, extending Fig. 3 (above right) to cover higher humidity levels. Based on these new results, we will more accurately determine whether a correction of the ISP-UNIVE record is necessary.

2) A first question would be what did you choose such low range for? If the effect of humidity was a goal of the study, then comparing the results of low (8-14000ppm) and recommended levels (17-22000ppm) would have indeed been relevant, but only if made on the same instrument.

The referee highlighted an important point. The level of 10,000-14,000 ppmv at Venice wasn't proposed as a target for comparing results at different humidity levels. Instead, it represents the maximum humidity level achievable with the setup proposed by ISP-UNIVE system, balancing melting speed, the collection of discrete samples via the Fraction Collector, online measurement and the need to supply a constant water volume to the CRDS instrument. The ISP-UNIVE system was a novel setup coupled with CRDS and will be further optimized based on the insights provided in this study to enhance future continuous isotopic measurements. As support to the reliability of the results presented by Venice, we are going to include a Section which include the impact of humidity levels based on the same instrument (as mentioned above).

This part will be revised in **Line 121-140**, as addressed in the response to Referee 2 (see comment 21)).

3) A second question is why did you decide not to apply any correction (i.e. humidity level calibration) on this particular dataset? If possible, a discussion regarding differences "with and without" correction can help the reliability.

We decided not to apply any correction to the data because, within the 10,000-14,000 ppmv range, the max observed variation - 0.1‰ for δ$^{18}$O and 1.2‰ for δD - is comparable to the confidence interval associated with 1-second integration times (~time of acquisition of Picarro instrument), as determined from the Allan Variance analysis. For this range of humidity, the deviation is such small that we are not able to separate it from the drift of the instrument. In addition, we highlight that both standards and samples were measured at the same humidity levels.
Applying a correction introduces additional uncertainties into the data, which would not be advantageous given the minimal correction that would result for this specific humidity range.

4) The study focuses on 2 French and 1 Italian laboratories, making "International" in the title a bit misleading. I would recommend a term like "Interlaboratory comparison" to better reflect the study.
Title: "Interlaboratory Comparison of Continuous Flow Analysis (CFA) Systems for High-Resolution Water Isotope Measurements in Ice Cores"

**Specific comments:**
**1/ Introduction:**
5) The authors present the CFA technique with respect to the traditional discrete sampling technique by focusing only on the reduced time of analysis. However, another aspect of CFA

is related to minimizing contamination, both by avoiding touching the sample, and the inner / outer channels at the melt-head. As all systems involved do not analyze only water isotopes, I assume that the melt-heads used feature both channels. Thus, this should be introduced.

We agree with this point.

Line 51-59: "This would take more than two years of discrete analysis if conducted with a single CRDS instrument. In contrast, with CFA-CRDS, operating at a melt rate of 2.5-3 cm min$^{-1}$, the same analysis can be completed in roughly three months, with the capability to process up to 10 meters of ice core per day. Additionally, the CFA offers the great advantage of providing - in parallel to the line for isotopic analysis a non-contaminated innermost melt water flow for further analysis. The innermost melt water flow is used for direct measurement, such as chemical analysis (trace elements, heavy metals, biomass burning tracers, etc.), and insoluble particle volume and distribution, and is simultaneously collected as discrete aliquots, greatly reducing the need for decontamination procedures in clean room. Despite its advantages, CFA-CRDS faces some technical limitations for isotopic analysis, one being the mixing of water molecules within the system, leading to signal smoothing (Gkinis et al., 2011)"

6) Lines 59-62: Dataset synchro and depth assignment... this comes out of the blue at this point. To me, these information belong to the method/data processing paragraph.

Line 59-63: "Additionally, measurement noise – referring to random fluctuations in the instrument' signal output – further limits the effective resolution, restricting the ability to retrieve meaningful climatic signals at high frequencies. "

Section 2.5

Line 192-216: "The isotopic raw data from the Picarro analyser, provided at acquisition time of ~1-s, are calibrated to the V-SMOW/SLAP and then are post-processed. The post-processing includes (i) converting the time to depth scale, (ii) filtering data affected by memory effects and artifacts and (iii) custom block averaged the data at resolution of 0.5 cm.
(i) The depth scale for the isotopic record is built through two common computational steps across all three laboratories. First, the timescale at MH is converted to depth using the measured ice stick lengths and the continuous recording of the encoder position. The encoder (located at the top of the ice sticks) records the melting of the cores at a frequency of ~1 Hz. Second, the arrival time at the CRDS, conductivity cell, and fractionation collectors is calculated based on the peristaltic pump flow rates. At ISP-UNIVE and IGE, the MH-to-CRDS delay is estimated through preliminary tests and corrected for pump rate changes. At LSCE, the MH-to-CRDS time for liquid phase is calculated using the continuously recorded flow rates and the volumes associated with each component of the CFA setup. Additionally, the delay time for gas phase transport to the CRDS is estimated based on isotopic steps from the SV, under standard operating conditions (specifically, $N_2$ and water flow rates). This gas phase delay is assumed to remain constant throughout the duration of the CFA run. Conductivity profiles plotted on the common depth scale help validated the processing. For LSCE, validation is based on the effective synchronization of three conductivity profiles. At IGE, where a single device is located prior the isotopic analyser, validation relies on aligning conductivity peaks – that correspond to transitions between individual ice sticks - with logged depths. Lastly, adjustments may be required between the actual stick length and the depth logged in the field, particularly for fragile and crumbling firn cores. Temporary blockages or stick collapses can introduce uncertainties when assigning depth to the isotopic profile. Whenever possible, such events should be documented and considered during the interpretation of the depth profiles.
(ii) Data at the beginning and end of CFA runs that are affected by mixing with pre- and post-circulated water are manually removed. At ISP-UNIVE, humidity drops – well below the typical work condition – are manually selected using a MATLAB graphical user interface and substituted by linearly interpolated values. (iii) Eventually, data are custom block-averaged at resolution of 0.5 cm."

**2/ Method:**

7) Line 104: Which Picarro is used where? Line 99 states Venice uses a l2140-i for discrete samples: Fig 2 shows a l2130-i for Venice CFA; Appendix A1 says the LSCE has a l2130-i; Appendix A2 says IGE has a l2140-i. Please introduce more and be consistent.

The previous version of the text did not clearly indicate that two different Picarro models were used at ISP-UNIVE for discrete (L2140-i) and continuous (L2130-i) measurements. We have revised the text as follows:

**Line 100-106**: "The discrete analysis were conducted at ISP-UNIVE using a Picarro L2140-i. The standards TD (Talos Dome) and AP1 (Antarctic Plateau 1) were used for calibration, while two vials of DCS (Dome C Snow) were analysed as controls. The accuracy of offline measurements was determined as the mean difference between control and true values of the STDs controls, with uncertainty represented by their SD. This yielding an accuracy of -0.01‰ for $\delta^{18}$O, -0.07‰ for $\delta$D, and -0.02‰ for d-excess, with corresponding uncertainties of ±0.07‰, ±0.4‰, and ±0.4‰, respectively.

The three CFA systems are coupled with CRDS Picarro-brand isotopic analysers: the L2130-i model at ISP-UNIVE and LSCE, and the L2140-i model at IGE."

8) Lines 138-139: *"The isotopic line includes an additional filter (identical to the LSCE one) and a conductivity device prior the analyser, maintaining a humidity levels between 17,000-21,000 ppmv."*
A filter and a conductivity device are used to maintain humidity levels...??!! please rephrase.

The entire section has been rewritten taking into account the comments from Referees 1 and 2 (see Comment 21).

9) ***Impact of humidity level:***
- See general comment.
- The data shown in Fig3 are results, thus should be presented in the Results section.
***Continuous Measurement Noise:***
Similarly, the results (Table 3, Fig 4) should be presented in the Results section.

We thank the referee for raising this point and we move Section 2.4.1 and 2.4.2 in the Results section.

10) ***Data Processing***
- Long paragraph (lines 191-215) with heavy sentences, not easy to read. Please rewrite it.

As described above, Section 2.5 has been rewritten

11) Complementary to the Yaxis isotopic data, having information regarding the X-axis (melt-rates plus standard deviation) should be given (in results section or at least in Appendix).

We agree with the Referee, and we believe that providing an indication regarding the X-axis could be a valuable addition. The revised version of the manuscript will include it for the Venice results, as an illustrative example in the Appendix.

12) Line 264-265: the quadrature difference was already mentioned, line 187.

Taken into account. This Section has been rewritten following the additional comments of Referee 2 (see comment 26)).

13) Lines 290-295: The author states that ISP-UNIVE-CFA data were not removed during data processing with intervals during which humidity level decreased down to 1200ppmv and 7850ppmv, respectively. But earlier (line 210), the authors stated removing data below 8000ppmv. This is either very strange and inconsistent, or the corresponding paragraph needs to be rephrased.

We have rewritten Section 3.2 as follows, incorporating the suggestions of Referee1 and Referee2:

**Line 285-307:** "We present the continuous $\delta$18O and d-excess records from ISP-UNIVE, LSCE and IGE in comparison with the discrete profiles (Fig. 7). For both ISP-UNIVE and IGE, the top 0.50 m of the first bag was removed due to complications encountered during the initial melting phase of the firn cores. These issues were related to the collapse of firn sticks at the melt head and the intrusion of

particles into the distribution line, which required cleaning with UPW. As a result, we show the PALEO2 data from the 12.5-16 m depth section. The CFA data, provided at 0.5 cm resolution after post-processing, was block-averaged at lower resolution by matching the discrete depth intervals (Fig. 7 a and c; original data in Appendix C, Fig. C1). The differences between the averaged continuous and discrete data are analysed using histograms of the differences at each depth point (Fig. 7. e and f), and statistical significance is assessed using a Kruskal–Wallis non-parametric ANOVA test. Differences with $p < 0.05$ are considered statistically significant. Overall, the variability in ice core $\delta 18O$ records, primarily at the decimetric scale, is comparable between the three CFA profiles and the discrete sampling, showing no statistically difference. In detail, the ISP-UNIVE-CFA shows difference of -0.01±0.26‰ (mean ± 1σ). Two data sections exhibit larger differences to the discrete profile, corresponding to depths of 13.85-13.96 m and 15.62-15.85 m (Fig. 7a, orange areas). The first interval involves ~15 cm of data removed and interpolated due to a humidity drop to 1,200 ppmv. The second interval shows a humidity fluctuation to 7,850 ppmv, slightly below the typical working conditions. This section has been retained. For the LSCE-CFA, the mean difference for $\delta 18O$ is slightly higher but remains non-significant, with a SD within the instrument's error (0.13±0.18‰ for $\delta 18O$, Tab. 5). The IGE-CFA results show a difference of -0.06±0.24‰ compared to the discrete data. The SD, similar to that of ISP-UNIVE but larger than that of LSCE, is primarily attributed to small depth scale shifts (Fig. 7b). For d-excess, the ISP-UNIVE and IGE show statistically difference from the discrete data of -0.78±0.64‰ and 0.88±0.48‰, respectively. These discrepancies are mostly attributed to calibration, as shown by the reduced difference after applying a calibration correction that aligns the mean difference to zero (Fig. 7d). In contrast, the LSCE record shows a non-significant difference of 0.03±0.55‰.

Overall, the good agreement between CFA and discrete record for both $\delta 18O$ and d-excess suggests that the LSCE data processing is the most reliable among the three CFA setups, which is why the entire PALEO2 core was analysed at LSCE (Sect. 4)."

14) Sentences like (line 295, 296: "*Although rare, such events occurred in the novel coupling of the CFACRDS at ISP-UNIVE, where achieving a highly consistent flow rate was challenging during initial analysis")* are not relevant for the reader.

Taken into account

**Additional points:**

Line 17: "instrument' signal output" → "instrument's signal output"

Taken into account

Line 29: low accumulation rates on the EAP of 20-50mm weq.yr-1. The accumulation rate at Kohnen station (DML plateau) is of 75 mm w.e.yr-1 (Wesche and others, 2016). Please correct.

Taken into account

Line 55: extend of this effect → extent of this effect

Taken into account

Line 88: remove δ2H

Taken into account

Line 130: "prior it enters" → "before it enters"

Taken into account

Line 205: synchronization instead of superposition

Taken into account

Line 344: Please delete sentences like "*We confirm that the primary advantage of the continuous method is its time efficiency"*

Taken into account

Line 398: TO clean the lines

Taken into account

---

## Author Comment (AC2)

Dear editor,

We would like to thank the two reviewers for their comments. We have been working toward a new version of the manuscript taking their respective comments into account. We include and number the reviewers' comments in **black**. The comments from Reviewer 1 are numbered from 1 to 14, and those from Reviewer 2 from 15 to 36. The referees' comments have been addressed individually, as requested by the journal. When both reviewers address the same topic, we indicate the corresponding reference number(s).

Our responses are in blue, and the modifications to the manuscript in red in this response file.

On behalf of all the co-authors,
Agnese Petteni

Review: An International Intercomparison of Continuous Flow Analysis (CFA) Systems for High-Resolution Water Isotope Measurements in Ice Cores by Petteni et al.

I. OVERVIEW

The manuscript *"An International Intercomparison of Continuous Flow Analysis (CFA) Systems for High-Resolution Water Isotope Measurements in Ice Cores"* by Petteni et al. deals with the comparison of three Continuous Flow Analysis systems developed for measuring the water isotopic composition ($\delta_{18}$O, $\delta$D) of ice core samples. The study focuses on comparison tests, providing quantitative evaluations of the resolution and precision achieved by the three systems. Example data from a firn core from Antarctica are used to support some of these tests.

This is a relevant contribution and it fits within the scope of the AMT journal. The manuscript is of good quality in terms of its methods and presentation; however, it lacks clarity and presents some important flaws and misconceptions, especially with respect to the diffusion and deconvolution parts. Therefore, I recommend publication after significant and major revisions are considered.

The manuscript has been partly re-written, refer to the specific comments for more details

II. GENERAL REMARKS

15) I think it is important for the authors to state early in the paper that this is an intercomparison using firn cores. While it is understandable that these systems can be used for ice cores, a significant part of the paper deals with sample diffusion effects, for which the porosity of the core sample is of immense importance. The term "international" should also be reconsidered and replaced with something more appropriate.

The flow of the paper is not entirely smooth, and there are sections from the Methods and Results that could, in principle, be bundled together. For example, Sections 2.6, 3.1, and 3.3 all deal with sample diffusion, signal attenuation, and spectral methods. The titles in their current form are vague and somewhat misleading.

We appreciate the feedback in identifying the aspects of the manuscript structure that may appear weak. We re-organize the text accordingly, with the following adjustments:

- We agree that the term "International" might not be appropriate in this context, and we are considering replacing it with "Interlaboratory Comparison of Continuous Flow Analysis (CFA) Systems for High-Resolution Water Isotope Measurements in Ice Cores".

- We will improve the organization of the Methods and Results sections, particularly by moving Sections 2.4.1 (Humidity level) and 2.4.2 (Continuous Measurement Noise) to the Results.

- The titles considered as misleading for section 2.6, 3.1, and 3.3, will be discussed more in details in comment 25).

- We agree with the referee that the results presented refer to firn cores, and that additional tests are necessary for ice core sections. We have revised the text accordingly.
  **Line 70-72**: "Power spectral density (PSD) analysis is used to quantify the effects of mixing on the signal and to determine the maximum resolution achievable by each system, as constrained by internal mixing and measurement noise. In this study, we focused on the analysis of firn by melting the upper part of the PALEO core. Although we present mixing lengths computed for various firn/ice transition pairings, firn-to-firn transitions could not be tested due to the unavailability of a mock stick with firn-like density. We also note that the results obtained may differ for deeper and denser ice core sections. While diffusivity at the melt-head is generally expected to be lower in denser ice due to reduced porosity, the transition from firn to ice is also influenced by changes in temperature and melt rate, which should be taken into account."

  **Line 190**: "We note that, due to the unavailability of a UPW mock stick with firn-like density, the $\sigma_{MH}$ values calculated in this study are based on tests involving different firn/ice

transitions. Specifically, a mean value derived from ice-to-ice, ice-to-firn, and firn-to-ice transitions is considered the best laboratory-based approximation of the effective mixing length. Although these values may vary depending on the density of the sticks when melted - due to capillary effects - this approach provides a reasonable estimate within experimental constraints as will be discussed in Section 3.1."

16) Some of the nomenclature used is atypical or incorrect. The authors use the term "mixing" extensively. This should be replaced with diffusion/dispersion/signal attenuation, as "mixing" describes a very different process governed by different models and mathematical frameworks.

Although we see the point of the reviewer, who considered the term "mixing" inappropriate, we think that our use may align well with the definition provided by Jones et al. (2017). In their work, Jones et al. (2017) distinguish between mixing and diffusion as follows:

- "The mixing length is a measure of $1\sigma$ displacement of water molecules from their original position in the solid ice sample. This is important for two reasons: (1) the system mixing length needs to be distinguished from the effects of diffusion occurring naturally in the ice sheet, and (2) based on the mixing length, system flow rates and the total mixing volume can be adjusted to prevent isotopic signal attenuation in ice cores from low-accumulation drill sites".
- "Isotopic mixing effects occur in the CRDS-CFA system. Possible contributors to the mixing effect include liquid mixing in tubing and the debubbler, liquid drag on tubing walls, vapor mixing downstream of the nebulizer, vapor interactions with two Picarro instrument filters (Mykrolis Wafer gard) prior to entering the laser cavity, adsorption of water molecules onto the laser cavity walls, and diffusional effects that can occur at any point in the CRDS-CFA system"
- "The standard deviation of the impulse response corresponds to the mixing length (often referred to as diffusion length in other publications), which defines the average movement of a water molecule in the time or depth domain relative to its original position in the ice sample or within a vial of water".

Anyway, we tried to better clarify the distinction between "mixing" and "diffusion" in the updated manuscript:

**Line 53-59**: "Despite its advantages, CFA-CRDS faces several technical limitations, one being the mixing of water molecules within the system, leading to signal smoothing (Gkinis et al., 2011). We use the term "mixing" for all the smoothing effects on the signal that occur within the CFA-CRDS system, including mixing occurring from the melt-head to the instrument cavity. On the other hand, we refer to "diffusion" for all the attenuation processes that happen naturally with in the firn. Overall, the transfer function of both processes is following the same equation (Eq. 1), but the length is usually longer for diffusion. The mixing length, influenced by technical setup and variations in core section density at the melt-head level due to capillary action, can differ significantly between systems. Consequently, isotopic values must be averaged over a depth interval equivalent to the mixing length to ensure accurate representation of the preserved climatic signal within the cores. (Gkinis et al., 2011; Jones et al., 2017a)."

We focus on presenting only the setup in Section 2.3, while we go into more detail in Section 2.4.3 to specifically address the mixing process.
**Line 125-126**: "The transport line typically includes a debubbler, which permits the release of air bubbles from the water stream as it passes through. "

**Line 178-180**: "The mixing in the CFA system attenuates the original signal preserved in the firn/ice cores, resulting in a smoothing of the isotope record (Gkinis et al., 2011). This effect, analogous to diffusion processes occurring naturally in the firn, arises from the displacement of water molecules from their original relative positions in ice matrix. However, a key distinction between CFA mixing and firn diffusion lies in their respective lengths: while diffusion typically acts at order of 6-8 cm, mixing is expected to occur over 0.7-1.5 cm (Gkinis et al., 2011; Jones et al., 2017a). Mixing occurs at multiple stages of the CFA setup, including at the melt-head due to water capillarity flow within the porous firn, in the debubbler, within the vaporiser chamber and CRDS cavity, as well as along the transport tubing in both liquid and vapor phase (Jones et al., 2017a)"

17) The same applies for the term "integrated" when referring to the averaging of the data over larger intervals.

We corrected the term "integrated" within the text:

**Line 286**: "We present the continuous $\delta^{18}O$ and d-excess records from ISP-UNIVE, LSCE and IGE in comparison with the discrete profiles (Fig. 7). The CFA data, provided at 0.5 cm resolution after post-processing, was block-averaged at lower resolution by matching the discrete depth intervals (Fig. 7. a and c; original data in Appendix C, Fig. C1). The differences between the averaged continuous and discrete data are analysed using histograms of the differences at each depth point (Fig. 7. e and f)."

18) An important aspect I struggled with while reading the paper was the sampling resolution of the datasets and the corresponding Nyquist frequency in the plots presenting data in the spectral domain. Please state clearly what your $\Delta x$ is, and clarify the 1-s sample acquisition time of the CRDS instruments. Typically, Picarro spectrometers export data at non-fixed time intervals. In all plots presenting spectral data, ensure that the Nyquist frequency/period is clearly indicated. Currently, the PSDs of the discrete data, for example, do not extend to 33.3 $m_{-1}$. What is the explanation for this?$_2$

This is correct. The discrete samples were cut with length of 1.5 cm; however, due to sample loss during the cutting process and uncertainties in the manual cutting, the final resolution achieved is approximately 1.7 cm. This explains why the PSDs of the discrete data extend to 3.4 $cm^{-1}$ rather than 3 $cm^{-1}$. Following this comment, we agree that it would be more appropriate to define the resolution of the discrete samples as 1.7 cm from the beginning of the manuscript. The main text has been rewritten as follows:

**Line 81-83**: "During the preparation of the ice sticks for the ISP-UNIVE measurements, discrete samples of 1.5 cm length were also cut. However, due to sample loss during the cutting process and uncertainties associated with manual cutting, the discrete samples resulted in a final resolution of 1.7 cm. The samples were stored frozen in PTFE bottles and analysed offline."

**Line 192-194**: We report changes done for Referee1 (comment 6)).
 "The isotopic raw data from the Picarro analyser, provided at acquisition time of ~1-s, are calibrated to the V-SMOW/SLAP and then are post-processed. The post-processing includes (i) converting the time to depth scale, (ii) filtering data affected by memory effects and artifacts and (iii) custom block averaged the data at resolution of 0.5 cm."

**Line 319-322**: "In this section, we conduct a PSD analysis on continuous and discrete results (Fig. 8) to assess the impact of mixing on continuous measurements and to determine the frequency limit at which the measurement noise begins to dominate over the signal. The continuous data has a post-processed resolution of 0.5 cm, while the discrete samples have a resolution of 1.7 cm."

19) Lastly, some of the central points of the paper regarding noise estimation and sample dispersion are based on calculations and data that are not clearly described and lack mathematical clarity. For example, the authors only present the fits to the impulse responses in Fig. 6 without specifying what exactly the reader is seeing.

We decided to keep separated the calculations from the results, and we described in the method Section 2.4.3 that the impulse response is determined by fitting a probability density function (PDF) to the first derivative of the Picarro response, described by normal Gaussian. The mixing length is then calculated as the standard deviation of this Gaussian. Instead, in Section 3.1 we present the corresponding results.
Regarding the description of the data, we agree and we further clarify Section 3.1 as follow:

**Line 255-273**:" Here, we present the mixing lengths of the three systems, evaluated from melt-head level (MH), which reflects the total mixing within the CFA-CRDS systems, and from the selector valve level (SV). The theoretical basis for these calculations is detailed in Section 2.4.3.

Due to the unavailability of a UPW mock stick with firn-like density, we tested different transition types: ice-to-firn, firn-to-ice, and ice-to-ice (Appendix B, Tab. B1). Although the mixing lengths may vary depending on the density of the sticks during melting, this approach provides the best estimate within the given experimental constraints, even if the values may be slightly underestimated for firn-to-firn case.

Figure 6a show an example of a normalised step function: firn-to-ice transitions are shown for ISP-UNIVE and LSCE, while an ice-to-firn transition is shown for IGE. Prior to normalization, the isotopic difference between ice sticks at MH (or liquid standards at SV) ranges from 40-50‰ for $\delta^{18}O$ and 380-400‰ for $\delta D$. The corresponding impulse responses are presented in Figure 6b. The mixing lengths from MH derived for each laboratory show no significant differences across transition types (Tab. B1). We therefore define $\sigma_{MH}$ as the mean value of all transitions, which will correspond to $\sigma mix$ in the back-diffusion approach presented in the following sections.
The mixing lengths ($\sigma_{MH}$, $\sigma_{SV}$ and $\sigma_L$) are summarized in Table 4. Since the values for $\delta^{18}O$ and $\delta D$ are very similar, we report only $\delta^{18}O$. The resulting mean $\sigma_{MH}$ values are 7.1 mm for LSCE, 11.3 mm for ISP-UNIVE, and 18.2 mm for IGE. Values expressed in seconds are converted in millimetres, using the average melting rate.”

**Line 275-276**: “Figure 6: a) Normalised transfer function and b) corresponding normal PDF impulse response function for ice steps at level of the CFA melt-head for $\delta^{18}O$. Firn-to-ice transitions are shown for ISP-UNIVE and LSCE, while an ice-to-firn transition is shown for IGE.”.

20) The Allan variance calculation also lacks explanation, and based on the shape and smoothness of the lines in Figure 4, it is quite clear that it has not been done correctly. Many of the claims regarding the mathematical treatment and the calculation of the transfer functions involved are incorrect, particularly in Sections 2.5 and 3.3. . The manuscript lacks a clearer mathematical foundation to support these claims. The same applies to the comparison between the CFA time series and the discrete data. The evaluation of the results is largely subjective.

These points are discussed more in detail in the following referee comments. See comment 23) for the mathematical approaches used in the Allan variance calculation and the transfer function, which have been further elaborated and integrated into the main text. Additional data for AV calculation will also be provided to support the plots presented in the manuscript, as mentioned in comments 1) and 22). Overall, we would like to emphasize that the Allan Variance is a well-established standard statistical method for assessing the stability of an instrument and don’t need a full mathematical development in the manuscript. Wikipedia, as example, has an exhaustive explanation of the Allan Variance (https://en.wikipedia.org/wiki/Allan_variance).

**III. MORE SPECIFIC ON THE VARIOUS SECTIONS**

*A. Experimental*

21) The manuscript can benefit from more clarity and depth relating to the description of the experimental part. This is very important for a submission to AMT. There is absolutely no information about the three vaporizers, a critical part of the system with respect to precision and achievable resolution. Are they all based on the capillary method? What is the diameter and the length of the capillary. What is the bore diameter of the tee split in the vaporizer? Also, information about the specifics of other parts of the system like the filter used. I would appreciate all the diagrams to be moved in the same section in the main text.
Since this paper assesses the performance of the systems based on firn analysis a subsection dedicated to the melters is essential. Capillary effect in the firn can significantly affect the dispersion of the sample and the choice of the melter design can have a big impact on these effects. Please provide the drawings of the three different melters. The same should apply for the design of the debubbler.

We agree with the suggestion to move all three diagrams to the main section of the manuscript, as this significantly improves the clarity in illustrating the differences in setup among the three systems. In addition, a description of the main characteristics of the vaporisers, melt-head and debubbler will be provided. The text is rewritten has follow:

Line 121-140:" The key differences between the three laboratories are summarised in Tab. 2. The novel ISP-UNIVE-CFA system (Fig. 2) is highly scalable according to the specific laboratory needs (Barbaro et al., 2022; Spagnesi et al., 2023). The melting unit is located within a vertical freezer. A conductivity device monitors the meltwater stream before it enters the small-volume 200 μl triangular flat cell debubbler. The debubbler has one inlet and two outlets: one for the debubbled meltwater and the other connected to the waste for the air bubbles and excess meltwater (~0.22 ml min-1).  The LSCE-CFA system (Fig. 2b) features a low-dead-volume glass debubbler with a volume of 430 μl, regulated by an automatic flow control to prevent overflow. The system includes a dust filter (A-107 IDEX stainless steel filter, 10 μm size: .189" x .074" x .254") and three conductivity devices.  The IGE-CFA system (Fig. 2c) supports a broader range of online analyses and includes a 1,000 μl low-dead-volume glass debubbler. Flow is manually regulated via pump adjustment, and the debubbler is connected to an open line and continuously monitored. Meltwater passes through a 180 μm dust filter and a longer distribution line. The isotopic line includes an additional filter (identical to the LSCE one) and a conductivity device prior to the analyser.

The three melt-heads used are similar, featuring a square cross-section with an inner and outer collection area separated by a 2 mm high triangular ridge, as the one described by Bigler et al. (2011). All the three vaporisers are similar and inspired from the capillary-based system described of Gkinis et al. (2010). LSCE and IGE vaporisers were both designed at Paris laboratory, while the ISP-UNIVE one was built at Ca' Foscari University of Venice. The stream is split from the incoming stream into a 50 μm inner diameter fused-silica capillary, and the rest goes into the waste line. The split takes place in a T split with a bore diameter of 0.5 mm. The sample micro-flow is injected in the oven (170°C at ISP-UNIVE and LSCE, and 190°C at IGE) , where it vaporises and mixes with dry air. At ISP-UNIVE a mass flow controller (Sensirion AG SFC6000D) is used to control the dry air.

The humidity levels at which the continuous analyses are performed are: 10,000-14,000 ppmv, 18,000-22,000 ppmv and 17,000-21,000 ppmv, respectively for ISP-UNIVE, LSCE and IGE. The lower range maintained by ISP-UNIVE represents the maximum level achievable with the nominal setup, balancing melting speed, discrete sample collection via the Fraction Collector, online measurement and the requirement to supply a constant water volume to the CRDS instrument.

In general, we do not aim here to isolate the specific impact of each individual system component, as this is beyond the scope of the present study. Rather, we provide a general overview of the factors that may contribute to mixing within the main liquid or vapor phases of the systems."

*B. On water concentration and Allan variance*

22) In Section 2.4.1, the authors conflate two distinct effects of water concentration on isotope spectroscopy. The first relates to the choice of water vapor concentration that yields optimal precision.

The second, as described in [2], concerns the dependence of the water isotope ratio signal on water concentration, which required a linear correction in that study. The selection of the 15,000–22,000 ppmv range in [2] pertains solely to the linearity of this dependence and the need for correction. However, the instrument examined in [2] is older and fundamentally different from the Picarro variants used in the present study.

The authors choose to show results only from the Venice system. As this is a technical intercomparison study, I believe results should be shown for all three systems and for both $\delta^{18}O$ and $\delta D$.

The referee is right. We will provide in the new version of the manuscript a new Fig. 3b, which will include extended time series of continuously measured UPW within the 10,000–20,000 ppmv

humidity range using the Picarro model 2130-i. These time series will allow us, on one hand, to quantify the variation in instrumental noise across different humidity levels, and on the other hand, to assess the relationship between humidity and the need for data correction.
Based on these new results, we will better re-define whether a correction of the ISP-UNIVE record is necessary. Given that any correction would introduce additional uncertainty into the dataset, this evaluation along with the noise calculated from AV is essential to justify whether a correction is beneficial within the observed humidity range. We won't be able to provide data for all the three systems, yet we reiterate here that the aim of this study is not to assess the performance of the analysers themselves, but rather to provide a general overview of the comparison between the three systems in the context of continuous isotopic measurements.

23) The Allan variance analysis appears problematic. Given the acquisition rate of the Picarro 2130 and 2140 models (2 Hz), the Allan variance curves should exhibit more high-frequency structure than the very smooth lines presented in Fig. 4. It seems clear that something else is being computed. Please consult [2, 4, 7, 8] for relevant plots and formulas. Additionally, a comment on how the authors transition from the non-fixed acquisition rate of the Picarros to a fixed timestep would be appreciated. Are the Picarros "pinged" at a constant interval via external control software, or is the data interpolated post-acquisition?
The manuscript mentions that the results in Fig. 4 are based on at least 2-hour injections of UPW. Why do the Allan variance curves stop at approximately 2000 seconds and not extend to at least 3600 seconds, which would be the expected upper limit for $\tau = t_{acq}/2$ [1]?[3]
It would be helpful to see the code used for this calculation, or at least a clear mathematical formulation. The $\delta^{18}O$ and $\delta D$ time series used in the Allan variance calculation should also be shown.

Picarro data provides outputs each 0.85 s for L2130-i model (ISP-UNIVE and LSCE) and at 0.72s for L2140-i model (IGE), the raw data are interpolated at 1-s post-acquisition. We implement this information in the text as follow:
**Line 161-162**: "For the ISP-UNIVE, LSCE, and IGE setups, we select a 1-hour interval with mixing ratios of 14,800±53 ppmv, 20,380±190 ppmv, and 18,100±105 ppmv, respectively. The raw data, provided each 0.85s for L2130-i models (ISP-UNIVE and LSCE) and each 0.72s for L2140-i model (IGE), are interpolated at 1-s post-acquisition".

We provide the continuous isotopic data for ISP-UNIVE in Appendix (Fig. A1 see below), and the formula used for the calculation rewriting the beginning of the Section 2.4.2 as follow:
**Line 157-159:** "To assess the stability and noise of the combined vaporiser and CRDS analyser, we calculate the Allan Variance (Allan, 1966) on isotopic time series (example of ISP-UNIVE time series is presented in Appendix Fig. A1). The time series are continuous measurement of UPW under constant humidity conditions that match those of CFA-CRDS analyses. The Allan Variance is computed taking a time series of size $N$. The data are divided into $m$ non-overlapping intervals, each containing $k=N/m$ data points. The acquisition time per data point is $t_i$, then the integration time for each interval is $\tau_m=k \cdot t_i$. The Allan Variance for a given $\tau_m$ is defined as:

$$\sigma^2(\tau_m) = \frac{1}{2m}\sum_{j=1}^{m}\left(\bar{\delta}_{j+1} - \bar{\delta}_j\right)^2$$

where $\bar{\delta}_{j+1}$ and $\bar{\delta}_j$ are the mean values of neighboring subsets $j$ and $j+1$. This method quantifies the time-dependent variance between consecutive intervals, making it particularly suitable for evaluating noise and drift in high-resolution continuous measurements."

[Figure]

Fig. R2. δ¹⁸O and δD results from a 2-hour continuous injection of UPW at ISP-UNIVE (~14,000 ppmv). (this figure will be included in the Appendix)

As the referee highlighted, for the calculation of the Allan variance presented in the first version of the manuscript, we effectively computed 2-hour injections of UPW but we selected only the last 1-hour to select a more stable range of data.

In Fig. R3 (below), we present a new calculation based on the full 2-hour range ($\tau = t_{acq}/2 = 3600$ s) for LSCE and UNIVE. For IGE, however, we had to exclude the first half hour, resulting in a usable portion of 1 hour and 30 minutes of continuous measurements suitable for Allan variance analysis. Nonetheless, even after modifying the selected time range, the SD$_{Allan}$ values at 2 s and 30 s-corresponding to 0.1 and 1.5 cm of melted core - remain unchanged.

We include the new figure in the manuscript and revise the text as follows:
**Line 159-163**: "The calculation is based on continuous UPW flow into the CRDS analyser under stable humidity conditions (Fig. 4, Tab. 3). For ISP-UNIVE, LSCE, and IGE, the mixing ratios are ~14,800 ppmv, ~20,000 ppmv, and ~18,100 ppmv, respectively. For ISP-UNIVE and LSCE we selected 2-hour of continuous data, while for IGE 1.5-hour time series was used. All systems show a decrease with a slope of N-1/2, characteristic of white noise, for at least $\tau$=250 s, indicating that precision improves with longer integration times."

More specifically: "The Allan variance analysis appears problematic. Given the acquisition rate of the Picarro 2130 and 2140 models (2 Hz), the Allan variance curves should exhibit more high-frequency structure than the very smooth lines presented in Fig. 4. "
It is not clear to us what the Referee means with high frequency structure.
If he means the long integration time structure (i.e. $\tau$=10^4 s): this part is extremely sensitive to the time series, as very few data points are available for the Allan Variance calculation (only 1.5–2 hours of data). Conversely, it the referee refers at high frequency (low integration time), we provide further support for the correct application of our Allan variance calculation by overlaying our results with those presented by Steig et al. (2014). This comparison show similar slopes to the ones of Steig (see Fig. R3 below), confirming the validity of our analysis.

Instead, regarding the absence of the higher-frequency graphical structure observed in the rightmost part of the plots by Steig et al. (2014) (Fig. R3) is simply an artifact related to the output mode of the integration times using the MATLAB function "allanvar". The output $\tau$ values are not evenly spaced but follow a logarithmic scale $\tau$ = [1, 2, 4, 8, 16, 32, 64, 128, 256, 512, 1024, 2048, 4096], resulting in this graphical difference.

[Figure]

Fig R3. (left) Steig et al. (2014) overlapped to our data, (right) our data.

24) For a technical intercomparison paper of this nature, one would expect a deeper analysis of the mechanisms underlying the significantly better precision observed in the Grenoble system. Water concentration level is not a plausible explanation, as it is comparable to that of the LSCE system. To state that "the IGE system exhibits higher precision, attributed to better instrument performance as indicated by Allan deviation" is a rather cyclical argument.

The better precision observed in Grenoble system may be related to the more recent Picarro instrument used at IGE (L2140-i model), compared to the L2130-i at ISP-UNIVE and LSCE. However, due to the large variability among Picarro instruments, explaining the better performance observed for the Grenoble instrument is too complex to delve into the technical details needed to explain the reasons behind this. We consider it beyond the scope of this study and won't investigate it further. We explain this in more detail in the main text, as follows:

**Line 166-169**: "Picarro L2140-i analyser used at IGE demonstrates higher precision than the L2130-i model used at LSCE, despite operating at lower humidity levels during the Allan variance assessment. In addition, the L2130-i analyser at ISP-UNIVE shows a precision comparable to that at LSCE, even though measurements were also conducted at lower humidity. These results suggest that precision and instrument noise are primarily determined by the analyser's intrinsic performance rather than the specific technical configuration of the CFA system. However, due to the considerable variability among Picarro instruments, a detailed explanation for the better performance of the L2140-i relative to the L2130-i lies beyond the scope of this study."

*C. Sample diffusion*

25) The diffusion of the sample in CFA systems and the resulting attenuation of the signal power is an important artefact that must be addressed. There are various approaches to this issue, one of which uses spectral methods and transfer functions estimated from impulse responses and/or step functions.
The manuscript presents some of these aspects in Sections 2.4.3, 2.6 (whose title should certainly be reconsidered), 3.1, and 3.3.
First, I find the term "mixing" misleading, as it technically refers to the blending of different compounds. Therefore, terms like diffusion, dispersion, and signal attenuation should be used throughout the manuscript.

As discussed in comment 16), the distinction between the terms "mixing" and "diffusion" is based on the definition provided by Jones et al. (2017), which we propose to rewrite more clearly at the beginning of the manuscript to ensure consistency throughout the text (see comment 16). Yet, we respectfully disagree with the assessment that mixing should not be used because the process is that

within the CFA line, liquid water is actively mixed, in the sense of different compounds of water with different isotopic composition are mixed with each other.

Based on the referee's comments, we kindly take the liberty to accept/decline the suggested section title changes, as outlined below:

- 2.4.3 Mixing in the CFA systems → Based on the definition by Jones et al. (2017), we refer to mixing as the combined effect of diffusion and dispersion of water molecules within the various components of the CFA system, which results in signal attenuation. This is distinct from the term diffusion, which we use to describe the natural process occurring in the ice. For this reason, we believe it is more appropriate to keep the original title.
- 2.6 Comparison Procedure → We revise it to: "2.6. Continuous and Discrete Isotopic Records Comparison"
- 3.1 Mixing lengths → We agree that the original title does not adequately reflect the content presented in the text. We have therefore revised it to: "3.1 Evaluation of Mixing Lengths from Impulse Response".
- 3.3 Spectral analysis → Also in this case, as suggested by the reviewer, we believe the following title to be more effective: "3.3 Impact of Diffusion and Mixing on the Signal".

26) In Section 2.4.3, the manuscript describes how $\sigma_L$ can be calculated, referring to [6]. However, based on the schematics of the three systems, $\sigma_{sv}$ is not equivalent to vapor diffusion as defined in [6]. Downstream of the selection valves, there are peristaltic pumps and filters in both the LSCE and Grenoble systems, all contributing to liquid-phase dispersion. The reason why $\sigma_{sv}$ is equivalent to $\sigma_{vapor}$ in [6] is that, in that setup, the tubing downstream of the selection valve is minimal and leads directly to a vaporization unit (nebulizer), eliminating the need for a pump. How does this significant detail affect the calculation of sample diffusion in the present study?

We fully agree with the reviewer's comment. In our case, we don't use the term $\sigma_{vapor}$ because it would not be used properly. In all the three systems, the $\sigma SV$ values reflect not only vapor-phase mixing but also include contributions from liquid-phase mixing in the section downstream of the selector valve. However, we emphasize once again that the main goal of this study is not to isolate the mixing effect of each specific CFA component, but rather to obtain a general understanding of the signal attenuation affecting the isotopic results. We write more in detail this part, as follow:

**Line 180-190**: "We assess the mixing effect occurring throughout the CFA systems and disentangle the main contributions in the phase upstream and downstream the selector valve. Two different impulse responses are evaluated. The first step function, generated at melt-head level (MH), involves the melting of two ice sticks in a row with different isotopic compositions. The second step function includes the mixing downstream the selector valve (SV) by switching between two isotopically distinct liquid samples. The impulse responses are characterised by fitting a probability density function (PDF) to the first derivative of the Picarro isotopic signal, described by the normal Gaussian:

$$f(x) = a_1 * exp\left(-\left(\frac{x - b_1}{c_1}\right)^2\right)$$

where $a_1$ is the amplitude, $b_1$ is the mean and $c_1$ is the standard deviation of the curve. Mixing lengths ($\sigma$) are defined as:

$$\sigma = \frac{c_1}{\sqrt{2}}$$

The $\sigma MH$ reflects the mixing of the entire CFA system, whereas the $\sigma SV$ represent the mixing downstream the SV. The latter includes the mixing caused by the presence of a peristaltic pump common to all the three systems and a mixing related to a filter prior the vaporizer for LSCE and IGE. To estimate mixing length upstream the SV, called $\sigma_L$, we calculate the root square of the quadrature difference between $\sigma_{MH}$ and $\sigma_{SV}$ (Jones et al., 2017a).

Previous studies used mock ice with varying isotope compositions to calculate σMH (Dallmayr et al., 2024; Jones et al., 2017a). In this study, the PALEO2 firn core (ρ = 0.58 g cm-3) is used, which has lower density and higher porosity compared to artificial ice (ρ = 0.92 g cm-3)."

27) Further on, in Section 2.6, the manuscript describes a modelling approach for the spectrum of the CFA data. The approach is problematic, as it assumes that CFA-induced diffusion adds power to the signal, represented by the term $P_1$. This is not physically possible. Diffusion does not add power—it only removes it.
Another important aspect missing from the analysis is the diffusion induced by discrete sampling itself. A sampling interval of 1.5 cm is roughly equivalent to a Gaussian transfer function with a diffusion length of 0.5 cm [5].

[Figure]

New Fig. 5 to include in the manuscript

We appreciate the reviewer's comments, which helped to highlight the weaknesses of this section and guided us in improving it as follows. We absolutely agree that diffusion does not add power. We modified the figure to really highlight that diffusion, and mixing were here to illustrate this removal of power. We included arrows and moved the diffusion and mixing label to really showcase that the signal is removed and not added. We have to point better out that the aim of Section 2.6 is to theoretically present the expected spectral effects of mixing and measurement noise associated with the CFA system, and predict their behaviour that will be later investigated in the power spectra of continuous measurements in Results.

To do so, we start from the discrete record, which we considered here as the best available approximation of the "true" signal.
   i.   The reviewer rightly points out that we do not account for the "diffusion induced by discrete sampling at a 1.7 cm sampling interval," which would affect period around 1.7 cm.
This type of diffusion is not physical but instrumental (Holme et al. 2018) - and limited to the individual bin (i.e., discrete sample) without propagating across adjacent bins - we consider its effect negligible since it should remain limited at period of 1.7 cm. Indeed, our scope is to observe the general features of CFA power spectra.

   ii.   Why there is anomalous power in the 5–20 cm period range in the discrete spectrum?

[Figure]

Fig. R4. to show how diffusion and mixing remove signal.

There are some effects that introduce noise in the 5-20 cm period, in parallel or after the diffusion effect. The fact that we are analysing a single ice core, in which stratigraphic noise induce horizontal aliasing could explain part of this excess signal in the 5-20 cm period range. Part of it surely is also measurement noise (around 0.01‰, leading to the PSD around $10^{-4}$ ‰² m at high frequency). Regardless of which process generated the signal, it is observed in the discrete samples, and it is removed in the CFA measurements. We argue that the mixing within the CFA is responsible for this. One of the argument is that the mixing length obtained from the spectrum matches with the mixing length calculated from the impulse method (Section 3.1 in the main text, Jones et al. 2017).

This is the reason why we observe the diffusion starts from ~50 cm (yellow) but do not completely erase the frequency in the 3-20 cm range. Otherwise, just in case of diffusion and measurement noise we should obtain a spectrum described by the red line (Fig R4):

$$P = P_0 \, exp\left(-k^2\sigma_{diff}^2\right) + \varepsilon_N$$

However, we do not investigate the different part of the climatic signal trapped in the spectra further, as it is not central to the aims of this study. For our purposes, we treat the discrete record (re-scale at 0.5 cm resolution - matching the resolution of the post-processed continuous data) as a sufficiently representative signal spectrum on which mixing and instrumental noise effects can be added and tested.

Schematic of depositional/post-depositional and measurement effects influencing the records of 12-16 m section of the firn core:

**Snow deposition**
- precipitation intermittency – white noise (it is the same for discrete and continuous measurements on the same core)

+

**Post-deposit. processes at the surface**
- noise in the record

+

**Diffusion and additional post-depos. processes that introduce noises (co-exist)**
- Diffusion attenuation starting from 50 cm period
- Post-depos. processes add power in 3-20 cm period

[Figure]

| CONTINUOUS ANALYSIS | DISCRETE ANALYSIS |
|:---:|:---:|
| + | + |
| **noise related to the cut of three different ice sticks, sections that can differ between each other** | **Instrumental diffusion** |
| • Responsible of part of variability between the continuous spectra | • (Holme et al. 2018, we consider it negligible) |
| + | + |
| **CFA Mixing** | **Instrumental noise** around 0.01‰ |
| • Mixing attenuates 3-10 cm | |
| + | |
| **Measurement Noise and Instrumental Diffusion** | |
| • MN Add white flat noise at high frequencies | |
| • ID negligible (Holme et al., 2018) | |

To sum up, this section serves as an illustrative example of the spectral effects expected from mixing and measurement noise. It is intended as a first-step conceptual approach that will support a more accurate isolation of these effects in the power spectral density (PSD) of the actual continuous datasets. As a result, we obtain a spectrum really similar to the ones presented in section 3.3.

**Line 218-245**: "The comparison between continuous and discrete isotopic records aims to highlight key technical differences in the CFA-CRDS setups and the operating procedures. Comparisons of profiles versus depth assess the agreement in calibration and depth scale attribution between laboratories. Additionally, the power spectral density (PSD) analysis – defined as the measure of signal's power content in the frequency domain - reveals system limitations caused by mixing and measurement noise. Before presenting the comparison in Sect. 3, we briefly introduce the PSD approach (Fig. 5), providing an idea of the mixing and measurement noise effects in the continuously measured signal. For this purpose, we consider ideally the discrete record as the best available approximation of the true signal preserved in the ice, limited only by discrete sampling resolution and uncertainties in the depth for sampling cut. The discrete spectrum is flat at frequencies around 100 m$^{-1}$ (white area), where the signal is dominated by precipitation intermittency and stratigraphic noise (Casado et al., 2020; Laepple et al., 2018). In contrast, attenuation begins at 50 cm$^{-1}$ (yellow area), consistent with diffusion effects (Johnsen et al., 2000). Firn diffusion lengths for the core sections analysed in this study (density of ~0.58 g cm$^{-3}$) have been estimated by previous studies to be approximately 6 cm (Johnsen et al., 2000; Laepple et al., 2018; Whillans & Grootes, 1985). The diffusion effect can be modelled by Eq. (1) (Johnsen et al., 2000):

$$P = P_o \, exp\left(-k^2 \sigma_{diff}^2\right)$$

where $\sigma_{diff}$ represents the firn diffusion length, and $k=2\pi f$, with f being the frequency. At higher frequencies (~3-20 cm$^{-1}$), a signal power is still observed. This noise likely arises in parallel with or after diffusion processes, and can be probably related to stratigraphic or other post-depositional processes. Indeed, we note that this spectral feature is preserved in both discrete and CFA records, and it appears attenuated in the latter due to signal mixing. However, the origin mechanism of this spectral power remains unclear and a comprehensive investigation of this phenomenon lies beyond the scope of the present study. Here, we aim to use the discrete dataset - resampled at 0.5 cm resolution to match the post-processed resolution of the continuous record - as a reference signal to which mixing and measurement noise are applied. To this end, we apply an additional Gaussian smoothing to account for the CFA system's mixing length ($\sigma_{mix}$) and incorporating the measurement noise ($\varepsilon_N$) determined for the online analysis, as described by Eq. (2):

$$P = P_1 \, exp(-k^2 \sigma_{mix}^2) + \varepsilon_N$$

As a result, the simulated spectrum diverges from the discrete spectrum at the frequency where CFA mixing begins to affect the signal, showing smoothing in the medium frequency range (>0.5 m$^{-1}$, orange area) and flattering at higher frequencies due to measurement noise (brown area). Measurement noise generates a flat spectrum at the frequency where the Signal-to-Noise Ratio (SNR) equals 1 (Casado et al., 2020), permitting to determine the frequency limit where meaningful climatic information can still

be retrieved as the point where the spectra of signal and noise intersect. Beyond this limit, noise dominates the signal, as the correlation between the record and the signal is defined as:

$$r^2 = \frac{SNR}{1+SNR}$$

At SNR = 1, a minimum significant correlation $r = \sqrt{0.5} \sim 0.71$ is reached. A similar behaviour is expected for the power spectral densities (PSDs) of the continuous records, allowing us to identify the maximum reliable frequency retained in the signal."

**Line 321-323**: "Diffusion, with a length of 6-8 cm in firn, begins to smooth the climatic signal at periods around 50 cm (as observed in the yellow areas of Fig. 8). In the range 20-50 cm, diffusion emerges as the dominant process shaping the spectra. At smaller scales (3-20 cm), however, additional mixing, characterised by mixing lengths of a few centimeters, further attenuates the power preserved in the discrete spectrum. This power is not only associated with instrumental noise but may be attributed to additional post-depositional processes occurring in parallel with or after diffusion, which will not be investigated in this study."

28) Throughout the manuscript, there is no information provided on the ice core site characteristics like temperature, accumulation and surface density. How do the authors estimate a firn diffusion length of 10–15 cm? Is this iceequivalent, or does it refer to firn density at the sampled depth?

The reviewer is right. We mistakenly referred to the firn diffusion length throughout the manuscript. The reference value we used - and which is consistent with the spectral analysis (red line in Fig. R4 above) -corresponds to $\sigma_{diff}$ = 6 cm. This length corresponds at the one reported by Johnsen et al. (2000) for the 12–16 m depth section of the Dome C core for $\delta^{18}O$ (see Fig. R5 below). We wrongly reported 12 cm in the previous version of the text, as 6*2 cm. We will revise the text accordingly, correcting the diffusion length to 6–8 cm for firn cores at this depth and density.

**Line 75-80**: "Four ice cores were drilled at the PALEO site (79°64'S, 126°13'E, mean annual temperature from ERA5: –46.5 °C) during the EAIIST on the Antarctic Plateau (Traversa et al., 2023; Ventisette et al., 2023). In this study, we focus on the PALEO2 firn core (18 m deep) to compare three CFA-CRDS systems. The full core was continuously analysed at LSCE in June 2023, while 4-m sections (12–16 m depth) were analysed at IGE in July 2023 and at ISP-UNIVE in January 2024. This 4-m interval, with an average density of ~0.58 g cm⁻³, was selected to explore the performance of the systems on low-density firn while maintaining sufficient structural integrity for handling and analysis in the cold room."

**Line 227-228**: "For the 12-16 m depth section of cores collected on the Antarctic plateau, such as the PALEO2 core (density of ~0.58 g cm⁻³), firn diffusion is estimated by previous studies to have a length of 6-8 cm for $\delta^{18}O$ (Johnsen et al., 2000; Laepple et al., 2018; Whillans & Grootes, 1985). Diffusion effect can be modelled by Eq. (1) (Johnsen et al., 2000): "

[Figure]

Figure 2:   Modeled firn diffusion lengths as a function of depth for both heavy molecule species $H_2^{18}O$ and $HD^{16}O$ at the indicated sites. Below the pore closure depth at some 60 m the diffusion lengths are only shortened by layer compression because the ice diffusion is negligible at these low temperature sites.

Fig. R5. (Johnsen et al., 2000)

29) In Section 3.1, the reader is presented with step functions and impulse responses, but without access to the underlying data or fits. For a technical publication like this, Fig. 6 should incorporate those elements. The information that the authors have used a sequence of firn–ice samples in various combinations is important.

In comment 19) we provide a more detailed description of the different firn/ice combinations used and presented. In addition, a new Fig. 6 will be implemented, incorporating the data used to calculate the step function.

30) Expected differences in diffusion characteristics due to capillary effects and firn porosity should be discussed.

See comments 15) and 19) for the description of the different transitions and the assumptions done relative to the capillarity effects and firn porosity. To further clarify this concept, we propose the following updated version in the Discussion:

**Line 359-365**: "Additionally, the CFA method is limited by mixing within the system which smooth the measured signal. In this study, we focused and presented results relative to firn cores. However, the impact of diffusivity may differ in deeper and denser sections of the ice core due to variations in ice porosity. Furthermore, changes in melt-head temperature and melt rate settings for denser core analyses may also influence the mixing impact. Therefore, additional tests are needed to accurately characterise mixing in deeper ice."

31) One of the most interesting results of the study—but insufficiently investigated— is why the LSCE system shows more diffusion downstream of the selection valve compared to the segment from the melter to the selection valve (8.6 s vs. 14.4 s, Table 4). The other two systems—and every system I am aware of—show the opposite behaviour. Additionally, the LSCE system does not appear to be fundamentally different from the others. This is something the authors should look into.

Between the three system, is ISP-UNIVE (and not LSCE, as mentioned by the referee) that shows more diffusion downstream of the SV compared to the segment from the MH to the SV, referred as $\sigma_L$ (see Tab. 4 below).

As suggested, we discuss this anomalous behaviour and revised this section as follows, discussing the $\sigma_L < \sigma_{SV}$ specifically of ISP-UNIVE setup.

**Line 267-273**: "Overall, LSCE system exhibits the smaller $\sigma_{MH}$, indicating the most efficient setup among the three systems evaluated. In contrast, IGE-CFA shows the highest $\sigma_{MH}$, with the dominant contribution arising from mixing in the liquid phase, as reflected by the higher $\sigma_L$. This is likely due to the presence of a high-volume debubbler and a longer distribution line, required to accommodate the higher number of online

measurements and discrete sampling operations performed by the laboratory. Notably, ISP-UNIVE shows more diffusion downstream of the selection valve (20.0s) than upstream (10.1s). This behaviour contrasts with the other systems presented in Table 4, including values for ice-to-ice transitions previously reported by Jones et al. (2017a) at the Institute of Arctic and Alpine Research (INSTAAR) Stable Isotope Lab (SIL) and by Dallmayr et al. (2024) at the Alfred-Wegener-Institut Helmholtz-Zentrum für Polar-und Meeresforschung (AWI). This higher $\sigma_{SV}$ observed in the ISP-UNIVE system is presumably attributed to the presence of a T-split before the vaporiser, which likely increases mixing. In addition, the relatively low $\sigma_L$ may result from the compact configuration of the system, there the melting unit is in a vertical freezer near the instruments, unlike located in cold rooms with longer distribution lines."

Table 4: $\delta^{18}O$ mixing lengths at melt-head level ($\sigma_{MH}$) and at selector-valve level ($\sigma_{SV}$) for the three different CFA-CRDS systems. The mixing length in the liquid phase ($\sigma_L$) is calculated as the difference in quadrature of $\sigma_{MH}$ and $\sigma_{SV}$. The mixing length expressed in seconds is converted in millimeters considering the average melting rate set at the three institutes. The 1SD are given in parenthesis. The mixing lengths are compared with Jones et al., 2017a at the Institute of Arctic and Alpine Research (INSTAAR) Stable Isotope Lab (SIL) and Dallmayr et al., 2024 at the Alfred-Wegener-Institut Helmholtz-Zentrum für Polar-und Meeresforschung (AWI).

|  | System | Melt rate (cm min$^{-1}$) | $\sigma_{MH}$ (s) | $\sigma_{MH}$ (mm) | $\sigma_{SV}$ (s) | $\sigma_L$ (s) | $\sigma_L$ (mm) |
|---|---|---|---|---|---|---|---|
| This work | ISP-UNIVE | 3 | 22.4 (2.0) | 11.2 (1.0) | 20.0 (0.6) | 10.1 | 5.1 |
|  | LSCE | 2.5 | 16.8 (2.2) | 7.1 (0.9) | 8.6 (1.4) | 14.4 | 6 |
|  | IGE | 3 | 36.4 (6.0) | 18.2 (3.0) | 16.8 (2.3) | 32.3 | 16.2 |
| From literature | INSTAAR | 2.5 | 17.4 (2.2) | 7 (0.9) | 9.4 (0.5) | 14.6 | 6 |
|  | AWI | 3.8 | 21.6 (2.4) | 13.6 (1.5) | 12.6 (1.8) |  | 4.5 |

32) Regarding Section 3.3 (Spectral analysis), I have several comments. First, it lacks a clear description of the mathematical foundation. The text describes the deconvolution step as an inverse Fourier transform, but it is4 not specified what exactly is being transformed. Do the authors construct a restoration filter? Is it optimized for measurement noise as in [3]? The text lacks both mathematical clarity and detail.

The filter is not optimized for measurement noise, as the latter is subsequently removed by custom block-averaging the signal at the maximum achievable resolution (one of the main goals of the study). We better present filter for deconvolving ice core data, as follow:

**Line 325 -329**: "While the effects of mixing can be corrected by applying back-diffusion to the signal, attempting this on frequencies dominated by measurement noise would result in an artificial amplification of that noise. However, as the primary objective of this study is to provide a straightforward approach for determining the resolution at which measurement noise begins to dominate the signal, the records will be custom-block averaged at that determined resolution. This process effectively removes measurement noise, additionally removing the amplification of noise that could result from back-diffusion process.

The filter used for deconvolving the mixing effect in isotopic time series, applies a back-diffusion method using a Gaussian kernel-based approach. Taking the time series and the nominal $\sigma_{mix}$ (Sect. 3.1) as input, for each data point in the time series, a Gaussian kernel is constructed based on the mixing length. The kernel is centred on the current point and extended to the surrounding points, with the width of the kernel determined by the diffusion length. The kernel is then normalized, and the values within the kernel range are weighted and convolved with the original data to produce a diffused value for each point. This smoothing process captures the effects of diffusion generating an artificial diffused record. Then, the inverse Fourier transform is calculated on the difference between original and diffused signals and is applied to the original signal to restore the higher frequencies."

33) There are also misconceptions regarding the influence of the various transfer functions (firn/CFA) on signal attenuation. A transfer function with a diffusion length of 15 cm has a much greater impact (several orders of magnitude) on cycles with periods of 3–20 cm (5–33

$m_{-1}$) than the CFA transfer function with a diffusion length of 1.5 cm. See plot below. So why do the authors claim that diffusion with a diffusion length of 10-15 cm primarily smooths the climatic signal over periods of 20-50 cm?

The referee is right in pointing out that this part was not clearly explained in the main text. What we want to explain is that the diffusion effect, which from the PSD analysis we observe starting to attenuate the climatic signal from 50 cm period, does indeed affect all higher-frequency spectra. However, we can define it as the "dominant effect" only within the 20–50 cm range. Indeed, in the period 3–20 cm, the mixing effect (lengths ~1.5 cm) also begins to play a role, and according to our results, mixing becomes the dominant process, further attenuating the power observed in the discrete record within this window (see new Fig. 5 above, which will be included in the revised version of the manuscript). We have better re-written these concepts above.

34) The authors claim a significant improvement in the 3–10 cm cycle range due to back diffusion correction, but no data are shown to support this. The data shown in Appendix D indicate the effect is negligible. Which is it?

We need to clarify that while back-diffusion effectively restores power at high frequencies, its impact on the overall ice core record is relatively limited. Therefore, applying this correction is not particularly meaningful in most cases - unless, for instance, one is specifically interested in investigating a past climatic event characterised by abrupt temperature changes.

**Line 329-334** : "The back-diffused profiles show significant improvement in the amount of signal across the 3-10 cm period range for all three CFA systems. The lack of signal for periods ranging from 10 to 20 cm in the LSCE profile, and to some extent in the ISP-UNIVE, is not corrected by this back-diffusion, which does not act at such frequencies. However, although we observe a correction of the high frequencies in the spectral domain, this adjustment proves to be negligible when comparing the back-diffused data with discrete samples along the depth scale (Appendix D, Fig. D1). This is because the signal is dominated by low frequencies -with approximately 1000 times more power at the 50 cm scale than at the 10 cm scale - and the restored high-frequency power remains relatively weak."

35) How is it possible that the measured signals lack cycles in the 10–20 cm range? A quick inspection of both the measured profiles and their power spectral densities reveals significant power in these frequencies. At the same time, a Wiener restoration filter for deconvolving ice core data with diffusion lengths of 13.4 and 16.4 mm is shown in Fig. 2 8 of [3]. It is clear that both these back diffusion filters—with values very similar to those in the current study—act extensively in this frequency range. Can the authors elaborate?

Clarifying these questions requires presenting the mathematics used—how is the restoration filter constructed, and what does it look like in the frequency domain?

The filter used is presented in comment 32) and implemented in Section 3.3

Differences in the variability in the ice – that can result in different variations between the continuous spectra - are not related to diffusion but may be related to artefact in the processing of the core: i.e. variability intra-core, cutting the ice core in sticks from three different sections of the cores for the three laboratories (see schematic above, comment 27)). However, the explanation regarding this aspect falls outside the scope of the present work, and won't be discussed more in details in this study. Our goal is to check here the impact of the CFA on the dataset.

*D. Discrete vs Continuous*

36) In the comparison between the produced time series, the terms "statistical difference" and "significant difference" are used. I believe it is important that the authors explain these terms and clarify what objective test they use for statistical significance. A sound normality test for the residuals between all the time series would greatly improve the manuscript. The Shapiro-Wilk and Anderson–Darling tests are some possible choices.

The reviewer is right on this point. We overlooked the fact that the Kruskall-Wallis nonparametric ANOVA test, used for the statistical analysis of the residuals in our study, was only mentioned in the table caption and not in the main text. The text will be revised accordingly as follows:
**Line 288-231**: "The differences between the averaged continuous and discrete data are analysed using histograms of the differences at each depth point (Fig. 7. e and f), and statistical significance is assessed using a Kruskal–Wallis non-parametric ANOVA test. Differences with $p < 0.05$ are considered statistically significant. Overall, the variability in ice core δ18O records, primarily at the decimetric scale, is comparable between the three CFA profiles and the discrete sampling, showing no statistical difference."

I believe that the manuscript needs extensive work in the review phase addressing these key points, therefore I will not add more minor comments in this review.

**REFERENCES**

[1] F. Czerwinski, A. C. Richardson, and L. B. Oddershede. Quantifying noise in optical tweezers by allan variance. Optics Express, 17(15):13255–13269–13255–13269, 2009.

[2] V. Gkinis, T. J. Popp, S. J. Johnsen, and T. Blunier. A continuous stream flash evaporator for the calibration of an IR cavity ring-down spectrometer for the isotopic analysis of water. Isotopes In Environmental and Health Studies, 46(4):463–475, 2010. doi: https://doi.org/10.1080/10256016.2010.538052.

[3] V. Gkinis, T. J. Popp, T. Blunier, M. Bigler, S. Schupbach, E. Kettner, and S. J. Johnsen. Water isotopic ratios from a continuously melted ice core sample. Atmos. Meas. Tech., 4(11):2531–2542, 2011. doi: https://doi.org/10.5194/amt-4-2531-2011.

[4] V. Gkinis, S. Jackson, N. J. Abram, M. Curran, T. Blunier, M. Halan, H. A. Kjær, A. Moy, K. M. Peensoo, T. Quistgaard, T. R. Vance, and P. Vallelonga. An 1135 year very high-resolution water isotope record of polar precipitation from the Indo-Pacific sector of East Antarctica. Australian Antarctic Data Centre, 2024. doi: http://dx.doi.org/doi:10.26179/ygeq-1a95.

[5] Christian Holme, Vasileios Gkinis, and Bo M. Vinther. Molecular diffusion of stable water isotopes in polar firn as a proxy for past temperatures. Geochim. Cos5 mochim. Acta, 225:128–145, 2018. doi: https://doi.org/10.1016/j.gca.2018.01.015. URL http://www.sciencedirect.com/science/article/pii/S0016703718300188.

[Figure]

Fig. 1. Diffusion transfer functions

[6] T. R. Jones, J. W. C. White, E. J. Steig, B. H. Vaughn, V. Morris, V. Gkinis, B. R. Markle, and S. W. Schoenemann. Improved methodologies for continuous-flow analysis of stable water isotopes in ice cores. Atmos. Meas. Tech., 10(2):617–632, 2017. doi: https://doi.org/10.5194/amt-10-617-2017.

[7] E. J. Steig, V. Gkinis, A. J. Schauer, S. W. Schoenemann, K. Samek, J. Hoffnagle, K. J. Dennis, and S. M. Tan. Calibrated high-precision 17O-excess measurements using cavity ring-down spectroscopy with laser-current-tuned cavity resonance. Atmos. Meas. Tech., 7(8):2421–2435, 2014. URL http://www.atmos-meas-tech.net/7/2421/2014/.

[8] P. Werle. Accuracy and precision of laser spectrometers for trace gas sensing in the presence of optical fringes and atmospheric turbulence. Applied Physics B-lasers and Optics, 102(2):313–329–313–329, 2011. doi: 10.1007/s00340-010-4165-9.

Fig. 2. Restoration filters

[Figure]

**References**

Aemisegger, F., Sturm, P., Graf, P., Sodemann, H., Pfahl, S., Knohl, A., and Wernli, H.: Measuring variations of δ 18O and δ 2H in atmospheric water vapour using two commercial laser-based spectrometers: an instrument characterisation study, Atmos. Meas. Tech., 5, 1491–1511, https://doi.org/10.5194/amt-5-1491-2012, 2012.

Christian Holme, Vasileios Gkinis, and Bo M. Vinther. Molecular diffusion of stable water isotopes in polar firn as a proxy for past temperatures. Geochim. Cos5 mochim. Acta, 225:128–145, 2018. doi: https://doi.org/10.1016/j.gca.2018.01.015. URL http://www.sciencedirect.com/science/article/pii/S0016703718300188.

E. J. Steig, V. Gkinis, A. J. Schauer, S. W. Schoenemann, K. Samek, J. Hoffnagle, K. J. Dennis, and S. M. Tan. Calibrated high-precision 17O-excess measurements using cavity ring-down spectroscopy with laser-current-tuned cavity resonance. Atmos. Meas. Tech., 7(8):2421–2435, 2014. URL http://www.atmos-meas-tech.net/7/2421/2014/.

---

## Editor Decision (ED1)

Review of "Interlaboratory comparison of Continuous Flow Analysis (CFA) Systems for High-Resolution Water Isotope Measurements in Ice Cores" by Agnese Petteni et al.

**Comments:**

Thank you for the revised manuscript, and my apologizes for the slow review.

The revised work has been significantly improved, and is now easier to read. However, there are still few points that I want to raise before accepting it:

**(A) ISP-UNIVE low humidity level:**

Ok, I agree that the accuracy of the measurement (i.e. standard deviation) calculated via the Allan variances is fully comparable, 'independently' of the water vapor level. But I am still concerned by the mean value itself (as I mentioned in my first revision, the previous Fig.3 (now Fig.5) shows a clear trend towards higher mean values, showing well that the result will be different between 8000 and 20'000ppm).

In section 3.4 "Discrete vs Continuous Data", lines 360 to 362, the authors state removing data at 1200 ppm, and retaining them at 7850 ppm. I would suggest to remove these 2 intervals, and rephrase the section accordingly.

- (B) A part from this remaining major concern, here are a few minor points:
- Line 226: "This is because, at these..." please rephrase.
- Lines 279 281: "Picarro 2140 ... also conducted at lower humidity". please also rephrase.
- Line 300: "For IGE setup, we relied on above tests confirmed the findings of Gkinis". Please rephrase
- Lines 320, 326: If I get it right, Tab. B1 should be **Table C1**
- Figure 6: On panel b), the change of slope is clearly occurring at 200s, normalized time. It would be good and consistent to have correspondence with panel a).
- Line 357: please insert a reference for the Kruskal-Wallis non-parametric ANOVA test.
- Line 362: 7850ppm, slightly below the typical working conditions. As the typical working conditions stated by the manufacturer are around 20'000ppm, this sentence made me smile, before raising my concerns again. cf comment (A).
- Line 401: Gaussian kernel-based approach. This also requires a reference.
- Line 504: ".. allowing the climate signal to be interpreted by reconstructing it at the highest retrievable resolution"

The highest retrievable resolution?? Higher than reconstructing climate signals by cutting/analyzing discrete samples at 1 mm resolution?? Please rephrase.

---

## Author Response (AR2)

Dear editor,

We would like to once again thank the reviewer for the valuable comments. We have addressed all the suggestions, which has allowed us to achieve what we believe is a much improved version of the manuscript. We are now looking forward to submitting the revised version. In this document, our responses are in blue, and the modifications to the manuscript in red in this response file.

On behalf of all the co-authors, Agnese Petteni

**Reviewer #1 (Remarks to the Author):**

Review pt. 2 of "An International Intercomparison of Continuous Flow Analysis (CFA) Systems for High Resolution Water Isotope Measurements in Ice Cores" by Agnese Petteni et al.

**General comments:**

- (A) ISP-UNIVE low humidity level:
- 1) Ok, I agree that the accuracy of the measurement (i.e. standard deviation) calculated via the Allan variances is fully comparable, 'independently' of the water vapor level. But I am still concerned by the mean value itself (as I mentioned in my first revision, the previous Fig.3 (now Fig.5) shows a clear trend towards higher mean values, showing well that the result will be different between 8000 and 20'000ppm)

The reviewer is absolutely right, it's important to evaluate the impact on the average isotopic composition with different humidity. We include the following text in the method and discussion sections:

Line 299: "Overall, we calculate an impact of humidity levels on the average isotopic composition of 0.015 ‰ per 1000 ppmv and 0.216 ‰ per 1,000 ppmv for  $\delta$ 18O and  $\delta$ D, respectively. Consequently, we opted not to apply humidity-level correction to the data for the range within which the measurements were performed."

Line 439: "The impact of humidity level on the measurements is around 0.015 % per 1000 ppmv for  $\delta$ 18O and 0.216 % per 1,000 ppmv for  $\delta$ D. Although these values are well within the uncertainty of the Picarro instrument, it is crucial to keep the humidity variation within a maximum range of 6,000 ppmv during daily CFA analyses to ensure that this effect remains negligible."

2) In section 3.4 "Discrete vs Continuous Data", lines 360 to 362, the authors state removing data at 1200 ppm, and retaining them at 7850 ppm. I would suggest to remove these 2 intervals, and rephrase the section accordingly.

We agree. The sentence has been rephrased as follows, and the figure has been modified accordingly:

Line 214: "At ISP-UNIVE, humidity drops – well below the typical work condition – are manually selected using a MATLAB graphical user interface and substituted by linearly interpolated values or removed when the depth intervals exceeds 10 cm."

**Line 360:** "Two data intervals, corresponding to depths of 13.85-13.96 m and 15.62-15.85 m are removed due to the drops of humidity at 1,200 ppmv and 7,850 ppmv, following the post-process procedure described in Sect. 2.5"

(B) A part from this remaining major concern, here are a few minor points:

Line 226: "This is because, at these..." please rephrase. Taken into account

Lines 279 - 281: "Picarro 2140 ... also conducted at lower humidity". please also rephrase. Taken into account

Line 300: "For IGE setup, we relied on above tests confirmed the findings of Gkinis". Please rephrase

Taken into account

Lines 320, 326: If I get it right, Tab. B1 should be Table C1 Taken into account

Figure 6: On panel b), the change of slope is clearly occurring at 200s, normalized time. It would be good and consistent to have correspondence with panel a).

Taken into account

Line 357: please insert a reference for the Kruskal-Wallis non-parametric ANOVA test. Taken into account

Line 362: 7850ppm, slightly below the typical working conditions. As the typical working conditions stated by the manufacturer are around 20'000ppm, this sentence made me smile, before raising my concerns again. cf comment (A).

Taken into account

Line 401: Gaussian kernel-based approach. This also requires a reference. Taken into account

Line 504: ".. allowing the climate signal to be interpreted by reconstructing it at the highest retrievable resolution"

The highest retrievable resolution?? Higher than reconstructing climate signals by cutting/analyzing discrete samples at 1 mm resolution?? Please rephrase.

**Taken into account**

Line 503: "Finally, the outcomes of this work gave the basis for the analysis of four PALEO cores analysed at ISP-UNIVE, LSCE, and IGE as part of the EAIIST (Traversa et al., 2023), enabling the determination of the maximum reliable resolution achievable with CFA for interpreting the climatic signal in ice cores."